# Towards In-context Scene Understanding 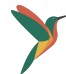

Ivana Balažević[*]  David Steiner[*]  Nikhil Parthasarathy[†]  Relja Arandjelović  Olivier J. Hénaff
Google DeepMind

## Abstract

In-context learning—the ability to configure a model's behavior with different prompts—has revolutionized the field of natural language processing, alleviating the need for task-specific models and paving the way for generalist models capable of assisting with any query. Computer vision, in contrast, has largely stayed in the former regime: specialized decoders and finetuning protocols are generally required to perform dense tasks such as semantic segmentation and depth estimation. In this work we explore a simple mechanism for in-context learning of such scene understanding tasks: nearest neighbor retrieval from a prompt of annotated features. We propose a new pretraining protocol—leveraging attention within and across images—which yields representations particularly useful in this regime. The resulting *Hummingbird* model, suitably prompted, performs various scene understanding tasks *without modification* while approaching the performance of specialists that have been finetuned for each task. Moreover, *Hummingbird* can be configured to perform new tasks much more efficiently than finetuned models, raising the possibility of scene understanding in the interactive assistant regime.

## 1 Introduction

In natural language processing (NLP), the pretrain-finetune paradigm has long been the dominant way of acquiring domain-specific knowledge and adapting a model's behavior to a particular task (e.g. question answering, natural language inference, summarization). More recently and predominantly due to the increase in model and dataset sizes, large language models have exhibited impressive, task-agnostic emergent capabilities [11, 37, 68], where a single model, given an appropriate prompt, can perform a wide range of downstream tasks without any change in its parameters.

While large-scale supervised and self-supervised pretraining in vision has yielded powerful encoders which capture useful semantics [15, 22, 31, 32, 35, 42, 43], applying these representations to solve downstream tasks has typically required bespoke decoders and end-to-end finetuning. The most readily applicable representations are trained for image-text alignment, enabling zero-shot classification [53] and image-based dialogue [2, 19, 80, 81], however these models are inherently limited by the coarseness of natural language outputs. Attempts have been made at casting fine-grained tasks (e.g. detection) as language modeling [17], but dense scene understanding tasks requiring millions of outputs do not lend themselves to this format. Indeed, deficiencies in fine-grained spatial understanding have been well documented in visual language models [36, 45, 64, 79].

In this work, we investigate the components required for in-context learning of scene understanding tasks, which we characterize along three axes: generality, data efficiency, and fast adaptation. To this end, we expand the well-known non-parametric nearest neighbor (NN) retrieval method [7, 9, 15, 75] to support a variety of dense scene understanding tasks. This retrieval-based decoding mechanism has the advantage of requiring no task-specific parameters or finetuning, thus enabling effortless adaption of standard encoders (e.g. ResNet [32] or ViT [22]) to any dense task of interest, as well as faster research iteration by allowing for simpler and more efficient model selection during pretraining.

---

[*]Equal contribution. [†]Current affiliation: NYU CNS, work done while interning at Google DeepMind. Correspondence to {balazevic, davidsteiner, henaff}@google.com.

37th Conference on Neural Information Processing Systems (NeurIPS 2023).

We further show that the NN scene understanding capabilities of canonically pretrained vision transformers (such as MAE [30] and DINO [15]) vary greatly, despite similar finetuned performance. We find two pretraining components to yield reliable gains: (1) a simple modification to the standard self-supervised pretraining protocol, termed *contextual pretraining*, which performs *attention across images* by updating the spatial representation of each image with features retrieved from a memory bank, and (2) a spatial *attention pooling* mechanism (as opposed to the more standard mean pooling or the `[CLS]` token), which computes *attention within an image* to summarize the (contextualized) spatial grid of features into a single image-level representation to be fed into the self-supervised objective. We showcase the benefits of this approach in a standard contrastive framework, demonstrating large gains in NN scene understanding over prior pretraining methods.

Finally we find that our model, named *Hummingbird* due to its fast adaptation properties: **(1)** yields general-purpose representations which perform well in non-parametric semantic segmentation and monocular depth estimation using NN retrieval, **(2)** approaches the performance of fully finetuned models on some tasks, and **(3)** is more data-efficient and faster to adapt to new tasks when equipped with NN retrieval, compared to other pretraining methods and decoding mechanisms. By adapting quickly and efficiently to new tasks specified on the fly, *Hummingbird* raises the possibility of vision systems providing general-purpose assistants with in-context scene understanding.

## 2   Related Work

**Retrieval-based perception.**    Non-parametric evaluation has a long history with roots in the exemplar theory of human cognition [3, 38, 50] and case-based theories of artificial intelligence [1, 58]. In computer vision, non-parametric methods combined with simple features such as SIFT [48] and HOG [21] saw early success in image classification [9], shape matching [7, 8, 59], scene recognition [66, 76], and image parsing [44]. Exemplar-SVMs [49] showcased the versatility of non-parametric methods by retrieving arbitrary meta-data (such as segmentations, geometry, even 3D models) from training examples. We leverage these insights with modern architectures and training paradigms coupled with dense retrieval.

**Retrieval-based training.**    To improve retrieval-based performance at test time, retrieval-based classifiers [69, 74] shape their representations for this task, enabling fine-grained classification from coarse supervision. While not explicitly training for it, DINO [15] witnessed NN classification abilities emerge from self-supervised training of vision transformers, enabling global retrieval tasks such as landmark recognition and copy detection. In [72], tracking abilities emerge after pretraining on a colorization task via retrieval from reference frames of a video. Retrieval has also been proposed as a means of enriching the positive pairs used in self-supervised contrastive learning [24]. These works differ from ours in that they encode and retrieve global representations of entire images, in contrast to the local inferences required by dense scene understanding tasks.

**Fast adaptation.**    A number of methods have tackled the problem of adapting to newly specified tasks, most often from the perspective of meta-learning. For example, matching networks [71] and MAML [26] learn to solve new classification and reinforcement learning tasks specified on the fly. Architectural innovations, such as image prompting [4, 39, 82] and adapter layers [27, 55] have also facilitated transfer to new image recognition tasks. While fast adaptation to dense scene understanding tasks has been less studied, image inpainting [6, 73] and VTM [41] have made progress in this direction, particularly in the low-data regime. These approaches differ from ours in that they achieve fast adaptation by training on related dense tasks and (in the case of VTM) adapt to downstream tasks with task-specific weight updates and learned similarity functions. In contrast, we maintain the simplicity of pure retrieval-based approaches by adapting to new downstream tasks without modifying any model parameters, and the generality of self-supervised approaches by learning representations from generic pretraining data with no dense annotations.

**Self-supervised learning.**    Methodologically, our representation learning method is most similar to self-supervised learning (SSL) techniques. Similarly to NLP, image-based SSL has witnessed great success in recent years, notably with the advent of contrastive methods [14, 15, 16, 23, 33], self-distillation [12, 18, 28], and masked auto-encoding [30]. Due to their conceptual simplicity, we base our method on standard contrastive baselines such as SimCLR [16] and MoCo [31]. Image-based SSL techniques have since been tailored to learning representations which transfer well to scene understanding tasks [13, 33, 70, 78], and although they have been shown to support zero-shot object discovery [34, 61], they generally still require task-specific decoders and end-to-end finetuning.

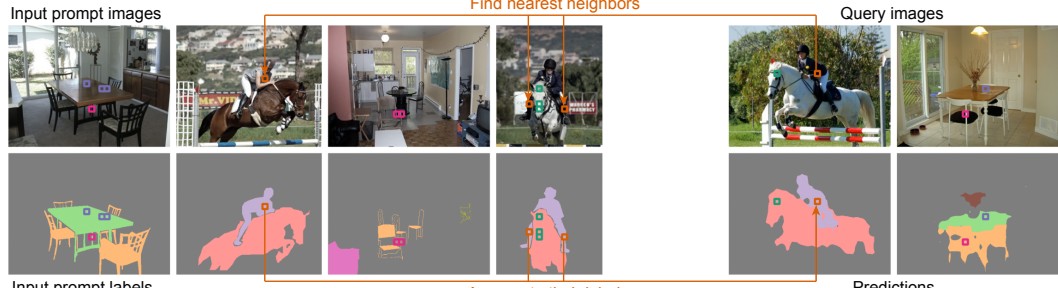

Figure 1: **In-context scene understanding with nearest neighbor retrieval.** On the left, we provide the system with a "prompt" of annotated images. On the right, we ask the system to describe new query images. The network computes dense features for each location and uses them to query features computed from the prompt. The labels associated with the nearest prompt features are then aggregated to make predictions about the query. Note that the system makes no assumptions about the nature of the labels, and as such can be used to solve a variety of different scene understanding tasks in-context. The nearest neighbors and predictions in this example are computed with our *Hummingbird* model.

## 3 Method

The following sections describe the retrieval-based scene understanding decoding protocol (Section 3.1), followed by the contextual pretraining method (Section 3.2) and the self-supervised (Section 3.3) and supervised learning objectives (Section 3.4). We use subscripts $\boldsymbol{x}_i$ to differentiate between representations and superscripts $\boldsymbol{x}^j$ to denote spatial locations within a representation.

### 3.1 Retrieval-based scene understanding

A general-purpose image representation should perform well across a variety of scene understanding tasks out-of-the-box, i.e. without modifying its parameters. To test whether a representation satisfies this condition, we extend the standard image-level nearest neighbor (NN) retrieval [7, 9] decoding mechanism to dense, patch-level retrieval (with patch size set to $16 \times 16$ across all models in this work). Given a prompt composed of training images from the downstream task and their corresponding labels $\{(\boldsymbol{x}_i, \boldsymbol{y}_i), i = 1, ..., N, \boldsymbol{x}_i \in \mathbb{R}^{H' \times W' \times C}\}$, our aim is to enable a pretrained image encoder $f_\theta$ to make predictions about a new image $\boldsymbol{x}$ from the test set. In tasks considered in this work, labels $\boldsymbol{y}_i$ are spatial maps of either class labels $\boldsymbol{y}_i \in \mathbb{C}^{H' \times W'}$ (e.g. for semantic segmentation, where $\mathbb{C}$ is the space of all classes) or scalars $\boldsymbol{y}_i \in \mathbb{R}^{H' \times W'}$ (e.g. for monocular depth estimation).

We encode each prompt image into a spatially flattened map $\boldsymbol{k}_i = f_\theta(\boldsymbol{x}_i) \in \mathbb{R}^{H \cdot W \times D}$, where a feature $\boldsymbol{k}_i^j \in \mathbb{R}^D$ at a spatial location $j$ is aligned with the local label $\boldsymbol{l}_i^j$ created by averaging the pixel labels $\boldsymbol{y}_i^j$ of a patch. We then sample a subset of features and local labels for each image, which form the keys and values of the memory bank $\mathcal{M} = \{(\boldsymbol{k}_i^j, \boldsymbol{l}_i^j), i = 1, ..., N, j \sim \mathcal{S}\}$ (see Appendix A.1 for details on the sampling distribution $\mathcal{S}$). In the following, we do not distinguish between entries from different images, and use a single integer $j$ to index into the memory bank: $\mathcal{M} = \{(\boldsymbol{k}^j, \boldsymbol{l}^j), j = 1, ..., |\mathcal{M}|\}$.

Given a test image $\boldsymbol{x}$, we form a representation $\boldsymbol{q} = f_\theta(\boldsymbol{x})$ and use each spatial feature $\boldsymbol{q}^i$ as a query to cross-attend over the memory bank with temperature $\beta$. The cross-attention weights are then used to combine the corresponding labels and form a local prediction $\hat{\boldsymbol{l}}^i$:

$$s^{i,j} = \frac{1}{\beta} \frac{\langle \boldsymbol{q}^i, \boldsymbol{k}^j \rangle}{\|\boldsymbol{q}^i\| \|\boldsymbol{k}^j\|}, \qquad \boldsymbol{a}^i = \operatorname*{softmax}_j(\boldsymbol{s}^i), \qquad \hat{\boldsymbol{l}}^i = \sum_j a^{i,j} \, \boldsymbol{l}^j. \tag{1}$$

Equation 1 defines the cross-attention operation as $\hat{\boldsymbol{l}}^i = \mathrm{CA}(\boldsymbol{q}^i, \boldsymbol{k}^j, \boldsymbol{l}^j)$. The final prediction $\hat{\boldsymbol{y}}$ is simply the concatenation of local predictions $\hat{\boldsymbol{l}}^i$ upsampled to the original image size via bilinear interpolation. As a result, nearest neighbor retrieval allows a simple image encoder to perform scene understanding tasks *without any decoders* or *parameter adaptation* (finetuning or otherwise) to the downstream dataset. The mechanism is also entirely agnostic to the format of the labels, enabling it to perform tasks as diverse as semantic segmentation and depth estimation.

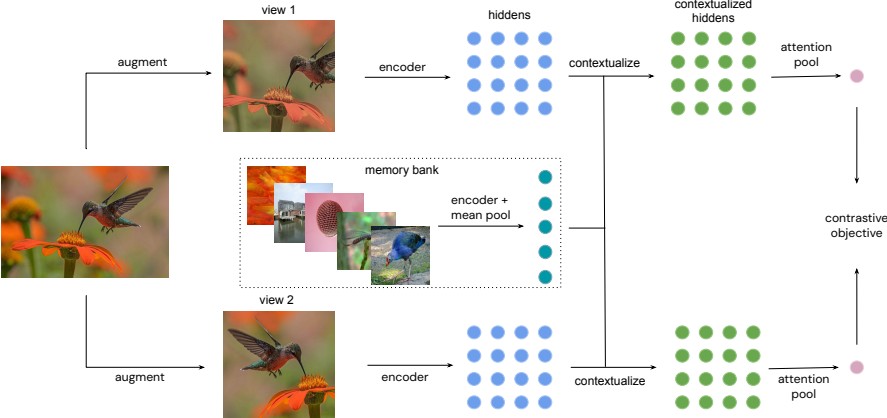

Figure 2: *Hummingbird* **model components.**

## 3.2 Contextual pretraining

Memory retrieval allows an image encoder to perform various tasks by combining labels of nearby examples. To ensure that a model will perform well in this regime, we propose to train it in a similar manner, by enforcing its representation to be expressed as a combination of representations of nearby examples. Over the course of training, we populate a memory bank $\mathcal{M}_p = \{(\boldsymbol{k}_i, \boldsymbol{v}_i), i = 1, ..., |\mathcal{M}_p|\}$ with spatially averaged keys and values computed from training images $\boldsymbol{x}_i$ from previous batches:

$$\boldsymbol{h}_i = f_\theta(\boldsymbol{x}_i) \in \mathbb{R}^{H \cdot W \times D}, \qquad \boldsymbol{k}_i = \frac{1}{H \cdot W} \sum_{j=1}^{H \cdot W} \boldsymbol{h}_i^j \in \mathbb{R}^D, \qquad \boldsymbol{v}_i = \phi_\theta(\boldsymbol{k}_i) \in \mathbb{R}^D, \quad (2)$$

where we use an MLP as the value head $\phi_\theta$ (see Appendix B for implementation details). We then form a representation $\boldsymbol{q} = f_\theta(\boldsymbol{x})$ of a new training image $\boldsymbol{x}$ and use each spatial feature $\boldsymbol{q}^i$ to attend over the memory bank and compute an update $\hat{\boldsymbol{v}}^i = \text{CA}(\boldsymbol{q}^i, \boldsymbol{k}_j, \boldsymbol{v}_j)$. Each feature is "contextualized" as $\boldsymbol{c}^i = \psi_\theta((1 - \lambda)\frac{\boldsymbol{q}^i}{\|\boldsymbol{q}^i\|} + \lambda\frac{\hat{\boldsymbol{v}}^i}{\|\hat{\boldsymbol{v}}^i\|})$, where $\psi_\theta$ is a linear layer and $\lambda$ a weighting parameter. The contextualized image representation $\boldsymbol{c} = g_\theta(\boldsymbol{q}, \mathcal{M}_p)$ is simply the concatenation of local features $\boldsymbol{c}^i$.

Note that the pretraining memory bank $\mathcal{M}_p$ is discarded at test time and differs from the test time memory bank $\mathcal{M}$ described in Section 3.1, allowing for straightforward comparison of our representations $f_\theta(\boldsymbol{x})$ to those trained without the memory bank.

## 3.3 Self-supervised objective

While contextual pretraining updates representations by attending across images, we hypothesize that learning to attend within images will also enable fine-grained predictions required by dense tasks. To that end, we train representations to locate the most distinctive part of an image using a combination of attention pooling and contrastive learning. Following [16, 28], we construct different views of unlabeled images $\boldsymbol{x}$ through random data augmentation $\boldsymbol{x}_1 \sim \mathcal{A}_1(\boldsymbol{x}), \boldsymbol{x}_2 \sim \mathcal{A}_2(\boldsymbol{x})$, see Appendix C.1. Each view is encoded as $\boldsymbol{h}_i = f_\theta(\boldsymbol{x}_i)$ and further contextualized with the mechanism described above as $\boldsymbol{c}_i = g_\theta(\boldsymbol{h}_i, \mathcal{M}_p) \in \mathbb{R}^{H \cdot W \times D}$ (see Figure 2). Following [52], we compute attention pooled representations $\hat{\boldsymbol{c}}_i \in \mathbb{R}^D$ using masks $\boldsymbol{m}_i$ derived from a lightweight attention module $a_\theta$, which we augment with an additional value head $\omega_\theta$. Pooled features are then used to compute projections $\boldsymbol{z}_i^\theta$:

$$\boldsymbol{m}_i = \text{softmax}_j(a_\theta(\boldsymbol{c}_i)), \qquad \hat{\boldsymbol{c}}_i = \sum_{j=1}^{H \cdot W} m_l^j \, \omega_\theta(\boldsymbol{c}_i^j), \qquad \boldsymbol{z}_i^\theta = p_\theta(\hat{\boldsymbol{c}}_i). \quad (3)$$

Finally, following [20, 28, 65], each view forms predictions $q_\theta(\boldsymbol{z}_i^\theta)$ about the other view's targets $\boldsymbol{z}_j^\xi$, which are computed with the same architecture and a different set of weights $\xi$ which vary more slowly (see Appendix C.2). The online weights $\theta$ are optimized using a standard contrastive loss:

$$\mathcal{L}_{\text{SSL}}^{ij}(\theta; \xi) = -\log \frac{\exp(q_\theta(\boldsymbol{z}_i^\theta) \cdot \boldsymbol{z}_j^\xi)}{\exp(q_\theta(\boldsymbol{z}_i^\theta) \cdot \boldsymbol{z}_j^\xi) + \sum_k \exp(q_\theta(\boldsymbol{z}_i^\theta) \cdot \boldsymbol{z}_k^\xi)}. \quad (4)$$

Table 1: **In-context scene understanding.** All models are pretrained on source data in a supervised or self-supervised manner, and applied to downstream datasets without modification. All downstream tasks are performed using a single mechanism, nearest neighbor retrieval. [†]indicates our reproduction of external work, all other models were evaluated using publicly available checkpoints.

| Method | Encoder | Params (M) | Dataset | Semantic segmentation | | Depth pred. |
| | | | | PASCAL ↑ | ADE20K ↑ | NYUv2 ↓ |
| --- | --- | --- | --- | --- | --- | --- |
| Supervised[†] | ViT-B | 86 | IN1K | 35.1 | 13.8 | .913 |
| DINO [15] | ViT-B | 86 | IN1K | 55.9 | 21.8 | .793 |
| MoCo-v3 [20] | ViT-B | 86 | IN1K | 37.2 | 14.6 | .771 |
| MAE [30] | ViT-B | 86 | IN1K | 6.6 | 3.3 | .981 |
| LOCA [13] | ViT-B | 86 | IN1K | 57.5 | 18.5 | .880 |
| *Hummingbird* | ViT-B | 86 | IN1K | 70.5 | 28.3 | **.718** |
| *Hummingbird++* | ViT-B | 86 | IN1K | **72.1** | **30.5** | .738 |
| | | | | | | |
| Supervised[†] | ViT-B | 86 | IN22K | 63.5 | 28.0 | 1.07 |
| MAE[†] [30] | ViT-B | 86 | IN22K | 9.8 | 4.2 | .968 |
| LOCA [13] | ViT-B | 86 | IN22K | 56.4 | 16.8 | .829 |
| *Hummingbird* | ViT-B | 86 | IN22K | 73.5 | 30.7 | .706 |
| *Hummingbird++* | ViT-B | 86 | IN22K | **76.2** | **34.1** | **.695** |
| | | | | | | |
| *Comparison across architectures:* | | | | | | |
| CLIP[†] [53] | NFNet-F6 [10] | 438 | ALIGN | 57.2 | 25.0 | .844 |
| Supervised[†] | NeXt-XL [46] | 1300 | IN22K | 58.9 | 25.5 | .791 |
| Supervised[†] | ViT-L | 307 | IN22K | 65.8 | 26.1 | .860 |
| MAE [30] | ViT-L | 307 | IN1K | 8.0 | 3.6 | .934 |
| LOCA [13] | ViT-L | 307 | IN22K | 59.5 | 17.6 | .912 |
| *Hummingbird* | ViT-L | 307 | IN22K | 76.9 | 35.0 | **.671** |
| *Hummingbird++* | ViT-L | 307 | IN22K | **77.3** | **35.8** | **.671** |

## 3.4 Retrieval-based supervised objective

Given the availability of large labeled datasets, and noting that correctly designed supervision does not necessarily hurt generalization [57], we explore the use of label-supervision for learning representations that perform well in dense NN retrieval. While supervision is typically added with a linear classifier atop average pooled features [32, 43, 62], we instead use it to constrain contextual pretraining and further align our training methodology with NN retrieval [69, 74]. Specifically, we expand the memory bank $\mathcal{M}_p$ to include the labels: $\mathcal{M}'_p = \{(\boldsymbol{k}_i, \boldsymbol{v}_i, \boldsymbol{y}_i), i = 1, ..., |\mathcal{M}'_p|\}$ and query it with attention pooled features $\hat{\boldsymbol{c}}_i \in \mathbb{R}^D$ (see Equation 3) to form predictions $\hat{\boldsymbol{y}}_i = \text{CA}(\hat{\boldsymbol{c}}_i, \boldsymbol{k}_j, \boldsymbol{y}_j)$. We then use the standard softmax cross entropy loss $\mathcal{L}^i_{\text{CE}}(\hat{\boldsymbol{y}}_i, \boldsymbol{y}_i)$, which added to the self-supervised objective of Equation 4, forms the total loss $\mathcal{L}^{ij} = \mathcal{L}^{ij}_{\text{SSL}} + \alpha(\mathcal{L}^i_{\text{CE}} + \mathcal{L}^j_{\text{CE}})$, with supervised weight $\alpha$. Note that the memory bank $\mathcal{M}'_p$ is only used during training and the added supervision relates to a global image classification task, not the downstream pixel-level tasks.

## 4 Experiments

We demonstrate the generality of *Hummingbird* representations through retrieval-based scene understanding on several downstream tasks (Section 4.1): semantic segmentation on PASCAL VOC [25] and ADE20K [83] with mean IoU (mIOU) as metric, and monocular depth estimation on NYUv2 [60] with root-mean-square error (RMSE) as metric. We further show that, in the low-data regime (Section 4.2) and when looking at adaptation speed (Section 4.3), *Hummingbird* with NN retrieval outperforms other pretraining techniques and decoding mechanisms, including end-to-end finetuning. Section 4.4 compares the performance of fully finetuned *Hummingbird* with prior work.

## 4.1 Retrieval-based scene understanding

We consider the performance of learned representations in the retrieval-based scene understanding setup described in Section 3.1 across architecture (ViT-B and ViT-L) and dataset (ImageNet-1k and

Table 2: **Data-efficient scene understanding.** After pretraining, models are adapted to downstream tasks on small amounts of data with end-to-end fine-tuning with a linear head (E2E FT) or with nearest neighbor retrieval (NN). $n$ refers to the number of images a fraction represents. All runs are averaged over five different seeds, with standard deviation of the order of 0.04 / 0.10% for NN and 0.36 / 1.35% for E2E FT on PASCAL / ADE20K. Each method is trained with ViT-B on ImageNet-1k.

| Method | Decoder | PASCAL ↑ | | ADE20K ↑ | |
|---|---|---|---|---|---|
| | | 1/128 ($n$=83) | 1/64 ($n$=165) | 1/128 ($n$=158) | 1/64 ($n$=316) |
| Supervised [67] | E2E FT | 41.8 | 53.8 | 10.8 | 14.3 |
| DINO [15] | E2E FT | 36.1 | 44.3 | 11.7 | 14.4 |
| MoCo-v3 [20] | E2E FT | 19.9 | 33.4 | 4.6 | 7.9 |
| MAE [30] | E2E FT | 34.2 | 44.1 | 8.2 | 12.2 |
| LOCA [13] | E2E FT | 40.1 | 53.9 | 11.2 | 15.5 |
| *Hummingbird* | NN | 50.5 | 57.2 | 11.7 | 15.1 |
| *Hummingbird++* | NN | **52.4** | **57.3** | **12.7** | **16.4** |

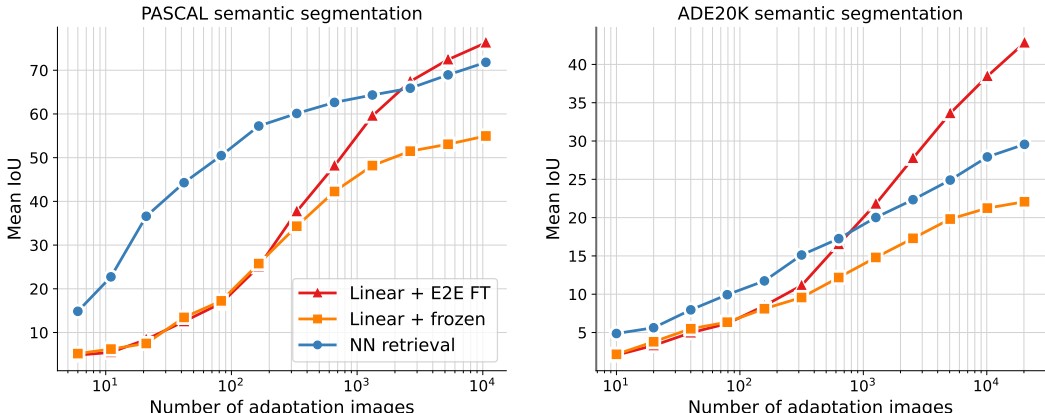

Figure 3: **Data efficiency of *Hummingbird*.** The model is evaluated with retrieval-based evaluation ("NN retrieval"), linear probing ("Linear + frozen"), or full finetuning ("Linear + E2E FT").

-22k [56]) scales, trained with supervision (*Hummingbird++*) or without (*Hummingbird*). Figure 1 shows an example prediction made by *Hummingbird* with a ViT-B encoder on PASCAL VOC.

The top part of Table 1 shows an apples-to-apples comparison of *Hummingbird* with existing methods for pretraining ViT-B encoders, where it outperforms all baselines by a large margin. We also note that *Hummingbird* scales well with increasing the dataset size from ImageNet-1k to ImageNet-22k, which does not hold for all other methods (e.g. MAE, consistent with [51]). Further, training with supervision is generally beneficial, particularly for semantic segmentation. For an ablation on the impact of retrieval-based supervision on performance, see Appendix D.5.

The bottom part of Table 1 contains the best performing methods across architectures, showing a performance increase for *Hummingbird* with encoder size. Note that results achieved by *Hummingbird* retrieval on PASCAL VOC and ADE20K, without any finetuning, approach the performance of methods fully finetuned on each of those tasks with specialized decoders (see Table 4).

## 4.2 Data-efficient retrieval-based scene understanding

In addition to adapting to downstream tasks with minimal (or ideally no) alterations to the model, a second ingredient for in-context learning is adaptation given only a limited number of examples.

We therefore evaluate the performance of *Hummingbird* retrieval in the low-data regime, and compare it with other decoding techniques: linear probing and end-to-end finetuning (Figure 3). For PASCAL VOC, NN retrieval outperforms the end-to-end finetuning for up to 1/8 of the data ($\sim$1300 images). For ADE20K the effect is less pronounced, however NN retrieval still exceeds end-to-end finetuning when given up to 1/32 of the data ($\sim$600 images). *Hummingbird* retrieval outperforms linear decoding

Table 3: **Fast adaptation to new scene understanding tasks.** After pretraining, models are transferred to downstream tasks with the full dataset, but a small amount of computation: 1 epoch. Models perform the task either with a linear classifier (Frozen), end-to-end fine-tuning (E2E FT) or with our mechanism for in-context scene understanding (NN).

| Method | Decoder | PASCAL ↑ Frozen | PASCAL ↑ E2E FT | ADE20K ↑ Frozen | ADE20K ↑ E2E FT |
|---|---|---|---|---|---|
| Supervised [67] | Linear | 61.5 | 66.3 | 27.6 | 15.1 |
| DINO [15] | Linear | 54.9 | 64.0 | 25.6 | 23.4 |
| MoCo-v3 [20] | Linear | 41.2 | 4.8 | 14.6 | 3.2 |
| MAE [30] | Linear | 20.1 | 42.5 | 8.3 | 7.9 |
| LOCA [13] | Linear | 61.9 | 62.9 | 25.4 | 14.6 |
| *Hummingbird* | NN | 70.5 | | 28.3 | |
| *Hummingbird++* | NN | **72.1** | | **30.5** | |

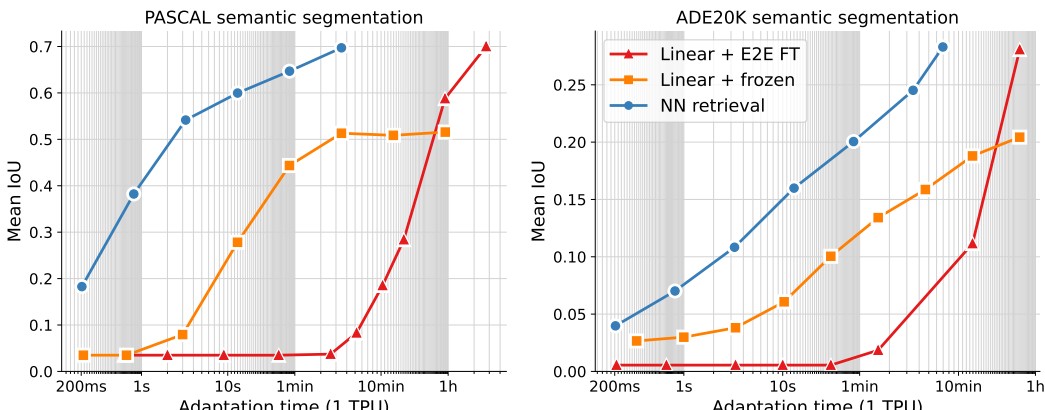

Figure 4: **Adaptation time of *Hummingbird*.** The model is evaluated with retrieval-based evaluation ("NN retrieval"), linear probing ("Linear + frozen"), or full finetuning ("Linear + E2E FT").

on top of the frozen encoder in all cases. These results show that given an appropriately designed encoder, NN retrieval provides a data-efficient alternative to end-to-end finetuning, and is strictly more expressive than linear decoding.

Second, we verify the generality of these findings by comparing *Hummingbird* retrieval to several other representation learning algorithms which transfer to the low-data regime with finetuning (Table 2, see Appendix D.1 for higher-data regime and additional analysis). For PASCAL VOC, *Hummingbird* with the NN retrieval decoder outperforms the end-to-end finetuned version of all other techniques for both 1/128 (83 images) and 1/64 (165 images) of the data, which holds for both the purely self-supervised *Hummingbird* and its supervised variant. For ADE20K, *Hummingbird* is competitive with DINO [15] for 1/128 of the data (158 images) and outperformed by LOCA for 1/64 of the data (316 images), whereas *Hummingbird++* outperforms all other models, demonstrating the benefit of retrieval-based supervision during pretraining. In summary, in the few-shot regime (e.g. ≤100 images) relevant for in-context learning, *Hummingbird* retrieval provides a compelling and robust alternative to end-to-end finetuning.

### 4.3 Fast adaptation to downstream tasks

While *Hummingbird* retrieval displays useful data-efficiency properties relative to fully finetuned methods, finetuning yields better performance when given access to the entire dataset. Yet even in this large-data regime, assistant systems must be quickly adaptable to new tasks. We thus evaluate the amount of computation required to reach good performance with the different decoding schemes from Section 4.2. All decoders are given the full training set and varying compute budgets. We titrate the amount of computation given to NN retrieval by partially populating the memory bank with fractions of the dataset. Figure 4 shows that 5 minutes (1 epoch through the downstream training set)

Table 4: **Scene understanding with end-to-end finetuning.** After pretraining, models are equipped with task-specific decoders and finetuned for that task on the entire downstream dataset. [†]indicates results are taken from [30], using UperNet [77] as the decoder. Results for all other baselines are taken from [13] and use the linear decoder from [63]. For ViT-L results, see Appendix D.4.

| Method | Encoder | Dataset | Fine-tuned accuracy (mIoU) | |
| | | | PASCAL ↑ | ADE20K ↑ |
| --- | --- | --- | --- | --- |
| Random | ViT-B | IN1K | 29.1 | 21.1 |
| Supervised [67] | ViT-B | IN1K | 76.1 | 47.3 |
| DINO [15] | ViT-B | IN1K | 74.1 | 44.1 |
| MoCo-v3 [20] | ViT-B | IN1K | 74.5 | 47.3[†] |
| BEiT [5] | ViT-B | IN1K+DALLE [54] | - | 47.1[†] |
| MAE [30] | ViT-B | IN1K | 75.0 | 48.1[†] |
| LOCA [13] | ViT-B | IN1K | 76.7 | 47.9 |
| *Hummingbird* | ViT-B | IN1K | 80.0 | 44.9 |
| *Hummingbird++* | ViT-B | IN1K | 81.2 | 44.9 |
| *Hummingbird* | ViT-B | IN22K | 81.6 | 46.9 |
| *Hummingbird++* | ViT-B | IN22K | **82.1** | **48.2** |

Table 5: **Ablation of pretraining components.** Effect of training with spatial attention pooling (as opposed to mean pooling or a `[CLS]` token) and memory contextualization ("Cont.") on performance. All models were pretrained with ViT-B on ImageNet-1k.

| Method | Pool. | Cont. | Semantic segmentation | | Depth pred. |
| | | | PASCAL ↑ | ADE20K ↑ | NYUv2 ↓ |
| --- | --- | --- | --- | --- | --- |
| MoCLR [65] | mean | ✗ | 38.6 | 4.9 | 1.01 |
| + cont. | mean | ✓ | 55.6 | 15.3 | .901 |
| + `[CLS]` | `[CLS]` | ✗ | 64.5 | 23.9 | .741 |
| + `[CLS]` + cont. | `[CLS]` | ✓ | 65.6 | 25.1 | .731 |
| + QK att. [52] + cont. | QK att. | ✓ | 68.7 | 26.3 | .728 |
| + QKV att. | QKV att. | ✗ | 68.0 | 27.4 | .742 |
| *Hummingbird* | QKV att. | ✓ | **70.5** | **28.3** | **.718** |

are sufficient to build a performant NN decoder (70% mIoU on PASCAL VOC, 28% on ADE20K). In contrast, given the same amount of time, end-to-end finetuning still exhibits performance near chance, despite benefitting from hyperparameter tuning of learning rates, weight decay, and warm-up length. While a linear classifier converges more quickly than finetuning, it saturates with a significantly lower performance than NN retrieval (50% mIoU on PASCAL VOC, 20% on ADE20K).

We also quantify these benefits in terms of relative convergence: on PASCAL VOC, NN retrieval reaches the performance of full finetuning after 3 minutes rather than 3 hours, and the performance of a linear classifier in 2 seconds rather than 3 minutes. For ADE20K, the speedups are smaller, but significant: 7 minutes rather than 30 minutes (relative to full finetuning), and 1 minute rather than 30 minutes (relative to the linear classifier). By making substantial gains in this near-real-time use case, we believe NN retrieval lays the groundwork for scene understanding in an interactive setting.

Table 3 compares *Hummingbird* retrieval to other models (equipped with linear or end-to-end finetuned decoders) in the fast-adaptation regime (i.e. when given a single pass over the full downstream dataset): *Hummingbird* retrieval outperforms all other pretraining techniques and decoding mechanisms on both PASCAL VOC and ADE20K.

### 4.4 Fully finetuned scene understanding

Although the primary focus of this work is on fast and effortless adaption to downstream tasks, for completeness, we include a comparison of fully finetuned *Hummingbird* with fully finetuned state-of-the-art models on the semantic segmentation task. We follow the finetuning protocol of MAE [30] and use UperNet [77] as a decoder. Table 4 shows that both *Hummingbird* and *Hummingbird++* are competitive with state-of-the-art when finetuned. Further analysis shows retrieval-based performance to be correlated with the finetuning performance (see Appendix D.2), paving the way for using retrieval-based evaluation as a model selection tool during training.

# 5 Analysis

**Ablating the pretraining components.** We perform an ablation of pretraining components required for adaptation to downstream tasks through NN retrieval in Table 5. We find attention pooling to yield superior performance compared to mean pooling or a `[CLS]` token. Both contextual pretraining and attention pooling separately lead to large performance improvements over a baseline MoCLR [65] model and best results are achieved when combining the two. Note that although spatial attention pooling was initially introduced in the context of video understanding [52], this work is the first to show its utility for downstream task adaptation in the NN retrieval setup. We further find that modifying it ("QK att") with a value head ("QKV att.") improves its performance across all tasks.

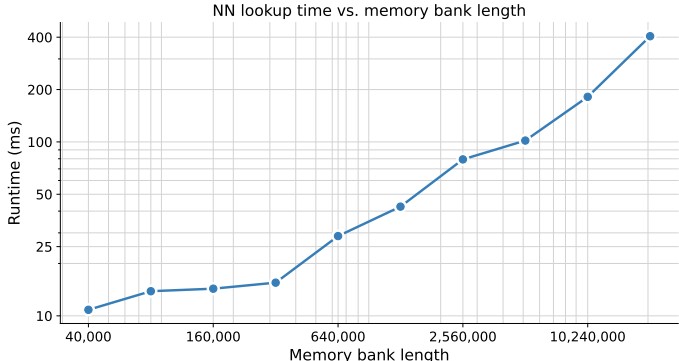

Figure 5: **Effect of memory bank length on nearest neighbor lookup at inference time.** Inference time is for a single image. Lookups were done on a single Nvidia A100 GPU.

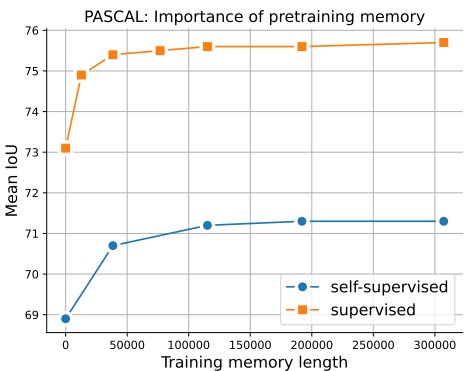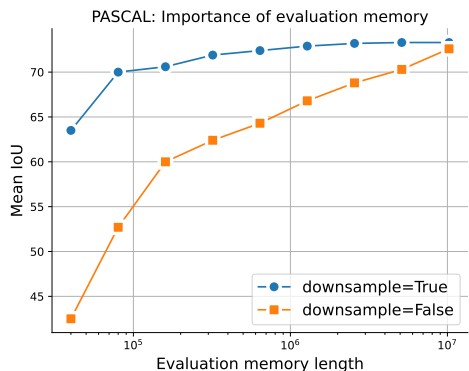

Figure 6: **Effect of the pretraining (*left*) and evaluation (*right*) memory length on performance.** All models were pretrained with ViT-B on ImageNet-22k. *Left*: Since the retrieval-based supervised objective is only defined for memory banks of non-zero length, for the purpose of this ablation we replace it with a simple linear classifier when $|\mathcal{M}_p| = 0$. *Right*: For downsample=False, we store representations of all patches into the memory bank. If downsample=True, we sample $|\mathcal{M}|/N$ patches per image ($N$ is the length of the downstream training set), allowing for greater diversity.

**Effect of evaluation memory length** $|\mathcal{M}|$**.** When transferring to downstream tasks with many training images (e.g. PASCAL VOC and ADE20K contain $\sim$ 10k and $\sim$ 20k images respectively, each image providing 100s of tokens), we see benefits of using large memory banks (e.g. $|\mathcal{M}|$ of the order of 1–10 million tokens, see Figure 6, right). Since this makes the cross-attention operation computationally intractable, we leverage powerful libraries for approximate NN search [29, 40] to limit cross-attention (Equation 1) to a small set of nearest neighbors for each query (e.g. $k = 30$, see Appendix A.2 for details, where we find increasing $k$ not to have significant impact on performance).

Figure 5 shows the relationship between evaluation memory length and the cost of the nearest neighbor lookup at inference time. For small-to-medium sized memory banks (0 to 1 million keys), the lookup cost is minimal ($\leq$ 30 ms), meaning the system is still fast enough to be used for real-time

applications, such as segmenting videos at 30 frames per second. When scaling to very large memory banks of 10 million keys or more, the scaling tends to be linear. However, the absolute performance is likely still suitable for most applications: with a memory bank size of 10 million, the overhead from NN lookup is only 0.2 seconds for dense tasks.

**Effect of pretraining memory length** $|\mathcal{M}_p|$**.** In contrast to retrieval-based evaluation, we find contextual pretraining to be remarkably $\mathcal{M}_p$-efficient: small memory banks (e.g. $|\mathcal{M}_p| = 40$k, see Figure 6, left for PASCAL VOC and Appendix D.3 for ADE20K) are sufficient to yield robust gains in retrieval-based scene understanding, adding a relatively small computational overhead to training the representation (e.g. +22% for $|\mathcal{M}_p| = 40$k). The module is agnostic to how the representation is trained and it benefits both self-supervised and supervised pretraining. Note that contextual pretraining is only present at training time and does not affect inference speed, and that pretraining and evaluation memory length are fully decoupled, allowing us to set them independently.

## 6 Conclusion

Inspired by impressive examples of in-context learning in language models, we investigate components necessary for in-context learning of dense scene understanding tasks in computer vision. To this end, we propose a simple non-parametric nearest neighbor retrieval mechanism—which is agnostic to the downstream task and requires no finetuning or specialized decoders—to serve as a general-purpose decoder which we use to evaluate models on semantic segmentation and monocular depth estimation tasks. We further propose *Hummingbird*, a pretraining method which benefits from attention across images (through contextual pretraining) and within an image (through spatial attention pooling) to produce image representations that can be easily configured to perform downstream tasks in a fast and data-efficient manner. By combining *Hummingbird* as the encoder with NN retrieval as the decoder, we take an important step towards in-context learning for dense vision tasks.

## 7 Broader Impact and Limitations

**Broader impact.** In laying the groundwork for scene understanding methods to be used in the interactive regime, our work could potentially benefit general-purpose assistants that are seeing rapid adoption. While these may enable a host of beneficial applications, they suffer from the biases and potential harms associated with visual language models and large language models more generally.

**Limitations.** Despite offering large relative improvements compared to finetuning and linear classification in the low-data regime, the absolute performance of *Hummingbird* when given less than 100 examples in the prompt is still far from perfect. To truly match the in-context learning abilities displayed in NLP, we would ideally need good performance from a handful of examples.

Further, given that retrieval-based scene understanding is task-agnostic, we leave expanding *Hummingbird* nearest neighbor evaluation to other tasks (e.g. object detection) to future work. Certain tasks, such as image rotation or image flipping, are not currently amenable to our framework as we assume a spatial correspondence between features and their labels. Explicitly post-processing to smooth outputs across patches was also not explored in this paper. Although the set of nearest neighbors and their weights vary smoothly as a function of the query image representation, it should be noted that this smoothness is dependent on the input prompt, since the memory bank needs to be sufficiently diverse and dense to allow a linear combination of neighbors to be expressive enough to cover all possible labels.

Finally, while we have showcased the benefits of attention within and across images in a contrastive framework, we defer adding them to more recent approaches using advanced data curation [51] and self-distillation [15, 84] to future work.

## Acknowledgements

We thank Daniel Zoran, Andrew Zisserman, Evan Shelhamer and João Carreira for their thoughtful feedback, Skanda Koppula and Mathilde Caron for their assistance in reproducing baselines, and Aäron van den Oord and Oliver Vikbladh for fruitful discussions at the inception of the project.

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

# A  Implementation details: retrieval based scene understanding

## A.1  How to store memories?

The memory bank only needs to be calculated once per dataset and can then be re-used for each of the images in the evaluation set. To populate the memory bank, each image in the dataset's training set (i.e. the "prompt") is encoded using the frozen backbone of the pretrained network to evaluate. We encode each of the training set images into a spatial map $\boldsymbol{k}_i = f_\theta(\boldsymbol{x}_i) \in \mathbb{R}^{H \cdot W \times D}$, where a feature $\boldsymbol{k}_i^j \in \mathbb{R}^D$ at a given spatial location $j$ is aligned with the local label $\boldsymbol{l}_i^j$ created by averaging the pixel labels $\boldsymbol{y}_i^j$ in that patch. These features $\boldsymbol{k}_i$ are then $L_2$-normalized.

When the memory bank length is not large enough to accommodate all features for all images, it is necessary to subsample and only store a subset of the features of each image. For a concrete example using ADE20K, training set images have a resolution of $512 \times 512$ which when encoded by a ViT-B/16 results in a $32 \times 32$ grid of features (i.e. 1,024 features per image). To store every feature from each of ADE20K's 20,120 training images would require a memory bank length of $20,120 \times 32 \times 32 = 20,695,040$. When using data augmentation to increase the number of training images, the required length is even higher.

Our subsampling strategy for semantic segmentation works as follows. We define the number of features to take per image as $n_{\text{features\_per\_image}} = \frac{|\mathcal{M}|}{|\mathcal{D}| * \text{num\_augmentation\_epochs}}$ where $|\mathcal{D}|$ refers to the number of images in the training dataset. We thus sample the same number of features for each training image. Rather than sampling this number of features per image from the grid uniformly, we attempt to sample the most salient features using a simple strategy: upweighting patches containing class labels that appear less frequently in the image. Following the notation of Section 3.1, let $\boldsymbol{l}^j$ refer to the label attached to the patch indexed by $j$ in the image and let $\mathbb{1}_{c \in \boldsymbol{l}^j} = 1$ if a given class $c \in \boldsymbol{l}^j$ and 0 otherwise. Then for each class $c$ we define $\kappa_c = \sum_j \mathbb{1}_{c \in \boldsymbol{l}^j}$ (i.e. a count of how many patches the class $c$ appeared in). We define a "class score" for each patch indexed by $j$ as $\text{class\_score}^j = \sum_{c \in \mathcal{C}} \kappa_c \cdot \mathbb{1}_{c \in \boldsymbol{l}^j}$. Finally, we take the $n_{\text{features\_per\_image}}$ from the spatial map $\boldsymbol{k}_i$ with the lowest final scores using

$$\text{final\_score}^j = (\text{class\_score}^j \cdot x) + (10^6 \cdot \mathbb{1}_{\boldsymbol{l}^j = \emptyset}) \tag{5}$$

where $x \sim \mathcal{U}_{[0,1]}$. The first term introduces some stochasticity into the sampling process and the second term deprioritizes locations that have no class label. The chosen features serve as the memory bank keys and their associated labels are the memory bank values.

The subsampling strategy used for depth estimation is simpler since there are no classes involved. We opted not to use data augmentation for this task making $n_{\text{features\_per\_image}} = \frac{|\mathcal{M}|}{|\mathcal{D}|}$. We first randomly order each patch in the image, then place all patches that contain no valid pixel labels after any patch with valid pixel labels, and then take the first $n_{\text{features\_per\_image}}$ from the list.

There are many possible alternative strategies for sampling the most salient patches within an image in the event that the memory bank length cannot fit every feature from every image. We leave exploration of these possibly better sampling strategies for future work because in general we found this technique to perform well and wanted to show that nearest neighbor evaluation does not require complicated, hand-crafted strategies but rather works well out of the box with a simple heuristic calculated per image. For a complete listing of the hyperparameters involved in building and retrieving from the memory bank, see Appendix A.2.

## A.2  How to recall memories?

After the memory bank has been populated as described in Appendix A.1, we sequentially make predictions for each image in the evaluation set. Evaluation was done on a single Nvidia A100 GPU per downstream task and takes approximately 15 minutes for PASCAL VOC, 25 minutes for ADE20K, and 30 minutes for NYUv2. Each image $\boldsymbol{x}$ is encoded as a grid of features $\boldsymbol{q} = f_\theta(\boldsymbol{x})$ and each of the features from this grid will serve as the query that we will look up the nearest neighbors for. We use the open-source ScaNN library [29] to perform the approximate nearest neighbor search efficiently. ScaNN natively provides the functionality to return both the top-k nearest neighbors for a given query as well as scores for the similarity that can be used as the attention logits. These scores

are then divided by a temperature scaling value before having a softmax applied to them to obtain the final attention values (see Equation 1).

Throughout the paper, we use ScaNN in asymmetric hashing (AH) mode as opposed to brute-force mode. We find that there is little to no negative impact on the evaluation from using approximate nearest neighbor search as opposed to a brute-force exact search, despite the approximate search being several orders of magnitude faster. We use cosine similarity ($L_2$-normalized dot product) as a distance measure throughout this work. We also attempted some experiments using squared Euclidean distance and found it to have no benefits to performance for any of the models evaluated.

Table 6: **NN retrieval hyperparameters.** Note that no training is involved with NN evaluation, hence there are no hyperparameters such as learning rates or training epochs.

|  | Section 4.2 | Everywhere else |
| --- | --- | --- |
| $|\mathcal{M}|$ (Memory bank length) | 20,480,000 | 10,240,000 |
| k (nearest neighbors) | 90 | 30 |
| Temperature | .1 | .02 |
| Augmentation epochs | 8 | 2 |
| ScaNN dimensions_per_block | 4 | 4 |
| ScaNN num_leaves | 512 | 512 |
| ScaNN num_leaves_to_search | 256 | 32 |
| ScaNN reordering_num_neighbors | 1800 | 120 |

Table 6 summarizes the hyperparameters used for NN evaluation throughout this work. For every section except for Section 4.2, we use a flat set of hyperparameters detailed in the "Everywhere else" column of Table 6. Because Section 4.2 is concerned with small subsets of the data (i.e. training on the order of hundreds of images), hyperparameter sweeps are extremely cheap to run and it is computationally fast to find nearest neighbors even with minimal approximations, hence we used a slightly different setup in this regime. In general, we found nearest neighbor retrieval to be surprisingly robust to the choice of hyperparameters, with temperature and reordering_num_neighbors being the most relevant to performance. The same set of hyperparameters were used for the semantic segmentation tasks (PASCAL VOC and ADE20K) as for the monocular depth estimation task (NYUv2), with the exception of the number of augmentation epochs (we did not use augmentations for depth estimation). For a complete description of the meaning of the ScaNN hyperparameters, please see `https://github.com/google-research/google-research/blob/master/scann/docs/algorithms.md`.

Table 7 details the parameters used for augmenting the training dataset for semantic segmentation tasks. Note that the augmentations used to augment the training set when evaluating downstream tasks differ from the augmentations used for creating different views of the same image during contrastive pretraining described in Appendix C.1. When augmentations are enabled, the image is first scaled between the minimum and maximum scale factor, from which a random crop is selected. Then photometric augmentations are applied independently with the probabilities and maximum intensities provided.

Table 7: **Evaluation augmentations.** Parameters used to augment the training dataset for semantic segmentation.

| Parameter | |
| --- | --- |
| Random crop probability | 1.0 |
| Minimum scale factor | 0.5 |
| Maximum scale factor | 2.0 |
| Brightness jittering probability | 0.5 |
| Contrast jittering probability | 0.5 |
| Saturation jittering probability | 0.5 |
| Hue jittering probability | 0.5 |
| Brightness adjustment max | 0.1 |
| Contrast adjustment max | 0.1 |
| Saturation adjustment max | 0.1 |
| Hue adjustment max | 0.1 |

# B  Implementation details: contextual pretraining

The contextual pretraining module takes as input a batch of image representations (i.e. queries) $q = h = f_\theta(x) \in \mathbb{R}^{B \times H \cdot W \times D}$ from the ViT encoder $f_\theta$, where $B = 4096$ is the batch size, $H = W = 14$ are the height and width of the spatial feature map and $D = 768$ for ViT-B and $D = 1024$ for ViT-L is the feature dimension. Keys and values for the contextualization cross-attention operation are entries of the memory bank $\mathcal{M}_p = \{(k_i, v_i), i = 1, ..., |\mathcal{M}_p|\}$, where keys $k_i$ are taken from previous batches by spatially averaging $h$ (see Equation 2) and values $v_i$ are obtained by applying a two-layer MLP $\phi_\theta$ to the keys, where we use batch norm after the first layer and the hidden dimension is set to 4096. Each feature $q^i$ of the image representation is then updated as $c^i = \psi_\theta((1 - \lambda)\frac{q^i}{\|q^i\|} + \lambda\frac{\hat{v}^i}{\|\hat{v}^i\|})$, where $\psi_\theta$ is a linear layer and $\|x\|$ is the $L_2$ norm. Preliminary analysis showed $\lambda = 0.2$ to work well across datasets, so we use it for all our experiments, with higher values $\lambda \geq 0.5$ degrading performance.

We populate the memory bank with all batch entries of ImageNet-1k / -22k at each step, using the representations from the target network. The memory bank is spread across 128 Cloud TPU v3 workers with 1200 entries on each TPU for ImageNet-1k (256 TPUs with 600 entries for ImageNet-22k), resulting in total memory length of 153,600.

# C  Implementation details: self-supervised pretraining

## C.1  Data augmentation

Each image is randomly augmented twice, resulting in two views $x_1$ and $x_2$. The augmentations are constructed as compositions of the following operations, each applied with a given probability:

1. random cropping: a random patch of the image is selected, whose area is uniformly sampled in $[0.08 \cdot \mathcal{A}, \mathcal{A}]$, where $\mathcal{A}$ is the area of the original image, and whose aspect ratio is logarithmically sampled in $[3/4, 4/3]$. The patch is then resized to $224 \times 224$ pixels using bicubic interpolation;

2. horizontal flipping;

3. color jittering: the brightness, contrast, saturation and hue are shifted by a uniformly distributed offset;

4. color dropping: the RGB image is replaced by its grey-scale values;

5. gaussian blurring with a $23 \times 23$ square kernel and a standard deviation uniformly sampled from $[0.1, 2.0]$;

6. solarization: a point-wise color transformation $x \mapsto x \cdot \mathbb{1}_{x < 0.5} + (1 - x) \cdot \mathbb{1}_{x \geq 0.5}$ with pixels $x$ in $[0, 1]$.

The augmented images $x_1$ and $x_2$ result from augmentations sampled from distributions $\mathcal{T}_1$ and $\mathcal{T}_2$ respectively. These distributions apply the primitives described above with different probabilities and different magnitudes. Table 8 specifies these parameters for the BYOL framework [28], which we adopt without modification.

Table 8: **Pretraining augmentations.** Parameters used to generate different views of a single image for contrastive pretraining.

| Parameter | $\mathcal{T}_1$ | $\mathcal{T}_2$ |
|---|---|---|
| Random crop probability | 1.0 | |
| Flip probability | 0.5 | |
| Color jittering probability | 0.8 | |
| Color dropping probability | 0.2 | |
| Brightness adjustment max | 0.4 | |
| Contrast adjustment max | 0.4 | |
| Saturation adjustment max | 0.2 | |
| Hue adjustment max | 0.1 | |
| Gaussian blurring probability | 1.0 | 0.1 |
| Solarization probability | 0.0 | 0.2 |

## C.2 Optimization

We pretrain the model for 300 epochs on ImageNet-1k or 100 epochs on ImageNet-22k using AdamW [47] with a batch size of 4096, split across 128 Cloud TPU v3 workers for ImageNet-1k and 256 Cloud TPU v3 workers for ImageNet-22k. Training a ViT-B / ViT-L for 300 epochs on ImageNet-1k takes roughly 21 hours / 53 hours, while 100 epochs on ImageNet-22k takes approximately 60 hours / 128 hours. We update the online parameters $\theta$ with a cosine learning rate schedule with a base learning rate of 0.001, weight decay of 0.1 and gradient clipping with a maximum norm of 1. We update the target parameters $\xi$ as an exponential moving average of the online parameters with a decay rate of 0.99.

Following [16] the projections and predictions in Equation 4 are normalized and rescaled such that their norm is equal to $1/\sqrt{\tau}$ where the contrastive loss temperature $\tau$ is equal to 0.1. When using additional supervision we set the supervised loss weight $\alpha$ to 0.25 for the supervised ViT-B trained on ImageNet-22k and $\alpha = 0.05$ for all other experiments.

# D Supplementary analysis

## D.1 Data efficiency

In Table 2 we compared *Hummingbird* with several leading representation learning techniques in the low-data regime. Here we provide the complete analysis from $1/128$ to 100% of the data, as well as results for our ViT-L model trained on ImageNet-22k to show the scaling properties of *Hummingbird*. Note that there is a difference between the experiments run here and those found in Section 4.4 of the main paper; that section uses an UperNet [77] decoder and this section uses a linear decoder for all of the finetuned rows in each table.

For PASCAL VOC (Table 9), *Hummingbird* performs very well not only in the low-data regime but in the full-data regime, with the apples-to-apples comparison (ViT-B self-supervised on ImageNet-1k) competitive with all other techniques even as the dataset fraction increases. This table also demonstrates the clear benefit of supervision as well as model-size and dataset size scaling—with only nearest neighbors (no finetuning), *Hummingbird++* trained on ImageNet-22k with a ViT-L backbone beats all of the other finetuned variants for every dataset fraction. *Hummingbird++* using a ViT-B and ImageNet-1k predictably lies in-between the other two models for every dataset fraction.

For ADE20K (Table 10), the same general trends from above hold. Backbone and dataset scaling are once again beneficial as *Hummingbird++* with ViT-L and ImageNet-22k training outperforms the other *Hummingbird* models, however this time the absolute performance relative to the finetuned competition in the high-data regime is less favorable since the end-to-end finetuned versions of other techniques start to outperform the nearest neighbors only ViT-L *Hummingbird++* at $1/16$ of the data.

Table 9: **PASCAL VOC data efficiency analysis.** After pretraining, models are applied to downstream tasks with the indicated fraction of the dataset size. Models perform the task either with end-to-end fine-tuning with a linear head (E2E FT) or with our mechanism for in-context scene understanding using nearest neighbors at evaluation time (NN). All fine-tuning runs are averaged over five different seeds. The metric reported is mean IoU (higher numbers are better). [†] denotes models trained on ImageNet-22k; all other models were trained on ImageNet-1k.

| Method | Decoder | Backbone | 1/128 | 1/64 | 1/32 | 1/16 | 1/8 | 1/4 | 1/2 | 1/1 |
|---|---|---|---|---|---|---|---|---|---|---|
| DeiT-III [67] | E2E FT | ViT-B | 41.8 | 53.8 | 63.1 | 67.7 | 70.7 | 72.2 | 73.4 | 75.2 |
| DINO [15] | E2E FT | ViT-B | 36.1 | 44.3 | 54.3 | 57.8 | 61.7 | 64.8 | 68.2 | 72.2 |
| MoCo-v3 [20] | E2E FT | ViT-B | 19.9 | 33.4 | 47.0 | 54.8 | 61.5 | 67.1 | 70.7 | 73.4 |
| MAE [30] | E2E FT | ViT-B | 34.2 | 44.1 | 53.0 | 58.7 | 62.7 | 67.4 | 70.8 | 73.5 |
| LOCA [13] | E2E FT | ViT-B | 40.1 | 53.9 | 63.1 | 67.8 | 70.7 | 72.8 | 74.4 | 75.5 |
| *Hummingbird* | NN | ViT-B | 50.5 | 57.2 | 60.1 | 62.6 | 64.3 | 65.9 | 68.9 | 71.8 |
| *Hummingbird++* | NN | ViT-B | 52.4 | 57.3 | 61.5 | 64.6 | 66.2 | 67.9 | 70.5 | 73.2 |
| *Hummingbird++*[†] | NN | ViT-L | **61.8** | **65.3** | **68.0** | **70.7** | **71.4** | **73.2** | **75.3** | **77.2** |

Table 10: **ADE20K data efficiency analysis.** After pretraining, models are applied to downstream tasks with the indicated fraction of the dataset size. Models perform the task either with end-to-end fine-tuning with a linear head (E2E FT) or with our mechanism for in-context scene understanding using nearest neighbors at evaluation time (NN). All fine-tuning runs are averaged over five different seeds. The metric reported is mean IoU (higher numbers are better). The results for other techniques between 1/32 and 1/1 are sourced directly from [13], the rest are reproductions. † denotes models trained on ImageNet-22k; all other models were trained on ImageNet-1k.

| Method | Decoder | Backbone | Fraction of dataset | | | | | | | |
|---|---|---|---|---|---|---|---|---|---|---|
| | | | 1/128 | 1/64 | 1/32 | 1/16 | 1/8 | 1/4 | 1/2 | 1/1 |
| DeiT-III [67] | E2E FT | ViT-B | 10.8 | 14.3 | 20.9 | 27.1 | 32.7 | 38.3 | 42.0 | 47.3 |
| DINO [15] | E2E FT | ViT-B | 11.7 | 14.4 | 18.4 | 24.5 | 29.5 | 35.2 | 39.5 | 44.1 |
| MoCo-v3 [20] | E2E FT | ViT-B | 4.6 | 7.9 | 17.7 | 25.2 | 30.8 | 36.5 | 40.7 | 45.4 |
| MAE [30] | E2E FT | ViT-B | 8.2 | 12.2 | 18.4 | 25.3 | 30.5 | 36.1 | 40.6 | 45.5 |
| LOCA [13] | E2E FT | ViT-B | 11.2 | 15.5 | 22.2 | **30.0** | **34.4** | **39.1** | **42.8** | **47.9** |
| *Hummingbird* | NN | ViT-B | 11.7 | 15.1 | 17.3 | 20.0 | 22.3 | 24.9 | 27.9 | 29.6 |
| *Hummingbird++* | NN | ViT-B | 12.7 | 16.4 | 18.9 | 21.5 | 24.0 | 26.8 | 29.9 | 32.0 |
| *Hummingbird++*† | NN | ViT-L | **16.6** | **20.5** | **24.0** | 27.4 | 30.2 | 33.1 | 36.0 | 37.8 |

## D.2 Correlation of NN retrieval and finetuning performance

In this section, we study the relation between NN retrieval performance and end-to-end finetuning. To that end, we collect 14 *Hummingbird* models trained with different architectures (ViT-B vs ViT-L), datasets (ImageNet-1k vs ImageNet-22k), learning objectives (self-supervised or with additional supervision), and training lengths. Figure 7 plots the performance of these models when equipped with NN retrieval decoders (x-axis) and fully-finetuned UperNet decoders (y-axis). For both PASCAL VOC and ADE20K semantic segmentation, performance using one decoding scheme is highly predictive of the other (Pearson's $\rho = 0.80$ for PASCAL VOC, $\rho = 0.89$ for ADE20K). As such, even in cases where NN retrieval underperforms end-to-end finetuning, it can still be used as a powerful diagnostic tool. As illustrated in Section 4.3, evaluating with NN retrieval is much simpler and faster than with end-to-end finetuning, even when using a linear decoder. End-to-end finetuning often requires sweeping over optimization hyperparameters and averaging across multiple seeds, making it unsuitable for online evaluation, whereas NN retrieval is $10\times$ less variable across runs and doesn't require any hyperparameter sweeps. As such NN retrieval can be used as an online evaluation that is highly predictive of performance obtained with more expensive finetuning protocols.

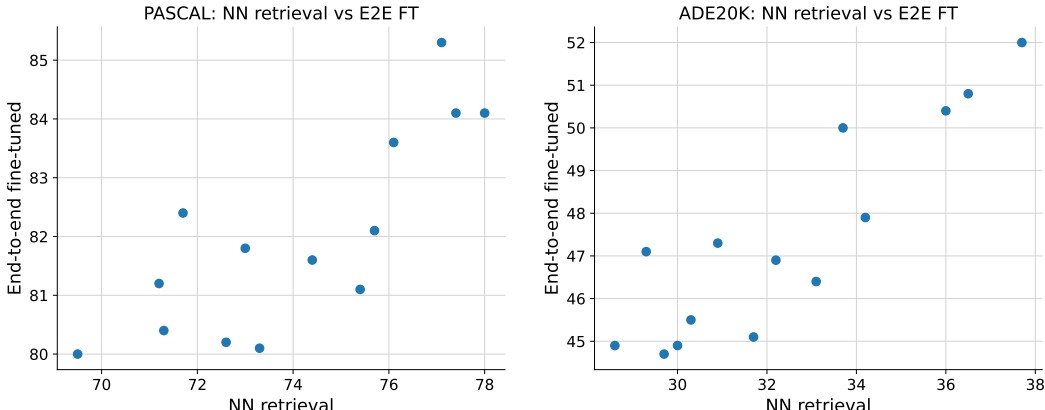

Figure 7: **Relation between NN retrieval and end-to-end finetuning performance.** We collect 14 models trained with different architectures, datasets, and learning objectives.

## D.3 Effect of pretraining and evaluation memory length for ADE20K

We include the equivalent of Figure 6 on the ADE20K dataset in Figure 8. Similar to what we observe for PASCAL VOC, we benefit from large memory banks at evaluation. Since the ADE20K

training set is roughly $2\times$ larger than that of PASCAL VOC, we also observe that sampling which features to store in the memory bank is more important than it is for PASCAL VOC (see Appendix A.1 on the details of the sampling procedure). Similarly, at training time, ADE20K benefits from larger pretraining memory banks than PASCAL VOC, with performance plateauing for memory banks larger than 200,000. Thus, we set the pretraining memory bank length to 153,600 in all our experiments (see Appendix B for details on contextual pretraining).

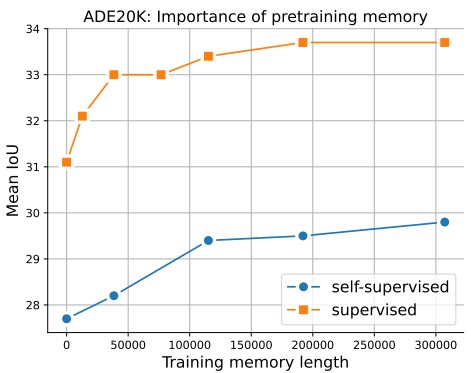 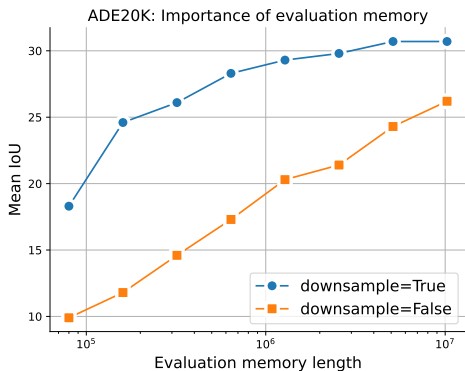

Figure 8: **Effect of the pretraining (*left*) and evaluation (*right*) memory length on performance of ADE20K.** All models were pretrained with ViT-B on ImageNet-22k. *Left*: Since the retrieval-based supervised objective is only defined for memory banks of non-zero length, for the purpose of this ablation we replace it with a simple linear classifier when $|\mathcal{M}_p| = 0$. *Right*: For downsample=False, we store representations of all patches into the memory bank. If downsample=True, we sample $|\mathcal{M}|/N$ patches per image ($N$ is the length of the downstream training set), allowing for greater memory bank diversity and thus superior performance than when downsample=False.

### D.4 Impact of encoder size on finetuned performance

We investigate the impact of encoder size in the finetuning regime in Table 11. We find that scaling the encoder from ViT-B to ViT-L is beneficial for both *Hummingbird* and *Hummingbird++*, where the self-supervised *Hummingbird* benefits slightly more from model scaling than its supervised counterpart. Note that this scaling study highlights the need for jointly scaling data and model size, as the best performing model overall is the one with ViT-L as an encoder trained on ImageNet-22k.

Table 11: **Impact of encoder size on finetuned performance.** After pretraining, models are equipped with task-specific decoders and finetuned for that task on the entire downstream dataset.

| Method | Encoder | Dataset | Finetuned accuracy (mIoU) | |
| --- | --- | --- | --- | --- |
| | | | PASCAL ↑ | ADE20K ↑ |
| *Hummingbird* | ViT-B | IN1K | 80.0 | 44.9 |
| *Hummingbird* | ViT-L | IN1K | 82.4 | 47.1 |
| | | | | |
| *Hummingbird++* | ViT-B | IN1K | 81.2 | 44.9 |
| *Hummingbird++* | ViT-L | IN1K | 81.8 | 47.3 |
| | | | | |
| *Hummingbird* | ViT-B | IN22K | 81.6 | 46.9 |
| *Hummingbird* | ViT-L | IN22K | 84.1 | 50.8 |
| | | | | |
| *Hummingbird++* | ViT-B | IN22K | 82.1 | 48.2 |
| *Hummingbird++* | ViT-L | IN22K | **85.3** | **52.0** |

### D.5 Importance of retrieval-based supervision for in-context scene understanding

We study the importance of retrieval-based supervised objective (see Section 3.4) on in-context scene understanding performance. We compare a model trained purely with the retrieval-based supervised objective ("Sup") with *Hummingbird* (purely self-supervised, i.e. "SSL") and *Hummingbird++* (both self-supervised and retrieval-based supervised, i.e. "SSL + Sup"). Results shown in Table 12 indicate

the necessity of the self-supervised objective for creating representations that are general and transfer well to downstream tasks.

Table 12: **Importance of retrieval-based supervision for in-context scene understanding.** All models are pretrained on source data and applied to downstream datasets without modification. All downstream tasks are performed using nearest neighbor retrieval. All models use ViT-B as an encoder.

| Method | Objective | Dataset | Semantic segmentation | | Depth pred. |
| | | | PASCAL ↑ | ADE20K ↑ | NYUv2 ↓ |
| --- | --- | --- | --- | --- | --- |
| Supervised retrieval | Sup | IN1K | 56.9 | 22.9 | .787 |
| *Hummingbird* | SSL | IN1K | 70.5 | 28.3 | .718 |
| *Hummingbird++* | SSL + Sup | IN1K | 72.1 | 30.5 | .738 |
| | | | | | |
| Supervised retrieval | Sup | IN22K | 69.3 | 30.3 | .739 |
| *Hummingbird* | SSL | IN22K | 73.5 | 30.7 | .706 |
| *Hummingbird++* | SSL + Sup | IN22K | **76.2** | **34.1** | **.695** |

