# OpenReview forum: "Towards In-context Scene Understanding"
_NeurIPS.cc/2023/Conference — NeurIPS 2023 spotlight_

### Official Review · Reviewer_35vE · 2023-07-05

**Soundness:** 4 excellent
**Presentation:** 4 excellent
**Contribution:** 3 good
**Rating:** 5
**Confidence:** 4

**Summary:**

The authors propose a self-supervised approach to adapt models learned via SSL to in-context learning via retrieval. To obtain the best performance, the authors propose to add attention pooling to encourage features attention (even though within an image) and supervised training for retrieval. Through an extensive number of ablations and empirical studies, the authors show the features produced via Hummingbird outperform other SSL approaches on segmentation and depth estimation.


**Strengths:**

1. The paper is well written and the presentation is very good.
2. The idea is to perform in-context via retrieval is very simple
3. The empirical results are effective and impressive compared to the baseline

**Weaknesses:**

Major:
1. There are a few potential limitations with the current approach to in-context learning. Since the main idea of in-context learning is the adaption to new tasks, it is important to explicitly mention what can and what cannot be done with Hummingbird.
2. With Hummingbird (no retrieval loss) there is a discrepancy between train time to test time

Minor:
1. Some missing RW.
2. L109 - slight inconsistency in notation going from HxWxD then using a single index for spatial location
Rating
3. What is the ViT-B patch size used? can you add it to Table 1/4?








**Questions:**

Major:
1. There are a few potential limitations with the current approach to in-context learning. Since the main idea of in-context learning is the adaption to new tasks, it is important to explicitly mention what can and what cannot be done with Hummingbird.

* The title implies that scene understanding, however, the retrieved features might actually be very local and specific. The features are retrieved independently of each other and there is no mechanism to consolidate the retrieved features into a whole scene output.
* Correspondence - with the current approach, the authors assume there are spatial correspondences between features and their associated labels. This is the case for segmentation or depth estimation but this is not necessarily the case for every possible task. E.g, if the task is to translate, crop and resize, rotate, or flip the image, etc, this assumption breaks.
* Retrieval approach might not result in smooth outputs. This is ok for segmentation but how would this work for style transfer?

I hope that the authors can explicitly clarify these in the final manuscript. On the one hand, the author's main claim is on segmentation and depth estimation, but on the other hand, in-context learning is very broad and therefore I think it should be stated explicitly what can be achieved and what not under this framework. I see this as the most important point, and I'll be willing to increase my score if this is addressed.

2. Discrepancy between train time to test time (attending within the image to attention across examples). I assume that with an end-to-end objective (perhaps similar to [2]), the gap between Hummingbird to Hummingbird++ can be diminished.


Minor:
1. Some missing RW.
* Recently retrieval-based a retrieval-based approach was proposed for choosing visual prompts [1]. To be fair, this work might be parallel research.
* Conceptually, the idea here is similar to a (now old) paper [2]. E.g., learning to attend to different features to colorize, which can later be used for segmentation. I think it is at least worth citing.
2. Why Hummingbird? Is there any connection from Hummingbird to the idea/method I'm missing?

[1] Zhang et al. "What Makes Good Examples for Visual In-Context Learning?". 2023.

[2] Vondrick et al. "Tracking Emerges by Colorizing Videos", ECCV 2018.


**Limitations:**

Please consider adding the following limitations (also stated above):
* The title implies that scene understanding, however, the retrieved features might actually be very local and specific. The features are retrieved independently of each other and there is no mechanism to consolidate the retrieved features into a whole scene output.
* Correspondence - with the current approach, the authors assume there are spatial correspondences between features and their associated labels. This is the case for segmentation or depth estimation but this is not necessarily the case for every possible task. E.g, if the task is to translate, crop and resize, rotate, or flip the image, etc, this assumption breaks.
* Retrieval approach might not result in smooth outputs. This is ok for segmentation but how would this work for style transfer?

---

> ### Author Rebuttal · Authors · 2023-08-09
>
> We thank the reviewer for their thoughtful comments.
>
> Major
>
> 1.
>
> * “The features are retrieved independently of each other and there is no mechanism to consolidate the retrieved features into a whole scene output.” We agree with the reviewer that the neighbors are retrieved independently for each feature, but the features themselves are strongly dependent on one another (e.g. since they may correspond to the neighboring parts of an image), which then leads to coherent predictions, as seen in Fig 1. This happens despite the lack of a more complex consolidation mechanism, as is becoming standard given powerful encoder networks (see e.g. linear probing with DINO v2 [1], or per-feature object detection in OWLViT [2]).
>
> * “Correspondence - with the current approach, the authors assume there are spatial correspondences between features and their associated labels.” We acknowledge that certain challenges like image rotation and image flipping do not work well with this framework and will add this to the Limitations Section. Other advanced tasks such as image translation could be more amenable to this setup, by leveraging more complex mechanisms like "representational algebra" as proposed in [3] and as is common in NLP (e.g. [4]).
>
>  * “Retrieval approach might not result in smooth outputs.” Indeed the discrete set of nearest neighbors will change as a function of the input. However the final prediction is a linear combination of their (continuous or discrete) labels, and the weighting will vary smoothly with the input representation. As a result, the NN predictions are as smooth as the underlying representation, particularly when using sufficient nearest neighbors (e.g. 30 or 90 in our experiments).
>
> * Style transfer: Somewhat surprisingly, NN (combined with a decoder) has recently been shown to be a useful component in SotA style transfer systems. See the diagram at the top of page 3 of Kolkin et al. ([5]) for a description of such a system. To do style transfer in our framework, the prompt would be composed of one or more style images, with the query image being the content image. However, as the reviewer correctly notes, with NN retrieval alone the image generated will not be perfectly smooth across patches. It would be necessary to add a decoding step as done in [5] (using our ViT backbone in place of VGG) to guide the content image towards the desired style.
>
>     We will include the summary of the points above in the Limitations Section and state more explicitly what the capabilities and limitations of the proposed framework are. Specifically, we note that our framework 1) relies on the dependencies between features to perform joint inference, and does not contain any notion of uncertainty when combining the predictions of multiple locations, 2) relies on a simple pairing of images and labels, and more complex relational reasoning (e.g. flipping labels) remains untested, and 3) relies on smooth features and dense memory banks to produce smooth predictions. We believe these points are the most important limitations of the approach.
>
> 2. The model is pretrained with both attention across images (through contextual pretraining) as well as attention within an image (through spatial attention pooling). At test time, NN evaluation makes predictions by linearly combining the labels of nearest neighbor examples. At training time, Hummingbird’s contextual pretraining mechanism aligns the representations with this kind of evaluation by expressing them as a linear combination of the representations of nearest neighbor examples. Even though adding retrieval-based supervision (i.e. Hummingbird++) brings pretraining even closer to how the model is evaluated, we find the self-supervised component to be crucial for maintaining generality and transferability of learned representations across tasks (see answer to Q5 of reviewer UEmV for an ablation and a more detailed discussion on the role of supervision).
>
> Minor
>
> 1. Thank you for the suggested references, we will cite them in the final version of the paper. To discuss each in more detail:
>     * [1] is based on choosing good images for the prompts to be used in Bar et al. (2022) "Visual Prompting via Image Inpainting", a paper we cite. [1]'s unsupervised method of choosing prompts picks images for the prompt such that they are most similar to the query image using a CLIP-based encoder. Intriguingly, their conclusion notes "CLIP would not be sufficient… A model that can better balance spatial and semantic closedness in feature space would be more ideal for visual in-context learning." Our localized and semantic features seem ideal for this use-case, making exploring their combination an interesting possibility.
>
>     * For [2], in addition to the difference in pretraining objective (colorization vs. contrastive learning) and backbone (ViT vs. CNN), other key differences are that their memory bank is much smaller ("four gray-scale video frames down-sampled to 256 × 256") and that the "model points within a single training example rather than across training examples". Thus the segmentation they describe is more applicable to tracking frames within a single video, rather than being general purpose.
>
> 2. We chose Hummingbird due to the model’s fast adaptation properties. Hummingbirds are known for their nimbleness (e.g. being able to fly both forward and backward and change directions quickly).
>
> Other
> * L109: Thank you, we will amend the inconsistency.
> * ViT-B (and ViT-L) patch size is 16x16 everywhere in the paper. We will include this in the paper.
>
> [1] Oquab, Maxime, et al. "Dinov2: Learning robust visual features without supervision." 2023
>
> [2] Minderer, Matthias, et al. "Simple open-vocabulary object detection." 2022
>
> [3] Eslami, SM Ali, et al. "Neural scene representation and rendering." 2018
>
> [4] Mikolov, Tomas, et al. "Efficient estimation of word representations in vector space." 2013
>
> [5] Kolkin, Nick et al. "Neural Neighbor Style Transfer." 2022

---

> > ### Comment · Reviewer_35vE · 2023-08-13
> > **Response to authors**
> >
> > Thanks for posting a rebuttal. Given that you plan to add the paragraph above to your limitation section, I changed my rating and I now support the acceptance of the paper.

---

> > > ### Author Response · Authors · 2023-08-15
> > > **Response to Reviewer 35vE**
> > >
> > > Thank you for increasing your rating and your support for the paper acceptance. Please let us know if you have any additional questions.

---

### Official Review · Reviewer_KiUe · 2023-07-05

**Soundness:** 4 excellent
**Presentation:** 3 good
**Contribution:** 3 good
**Rating:** 7
**Confidence:** 3

**Summary:**

This paper investigates in-context learning in computer vision. It proposes the Hummingbird model, which uses a pretraining protocol and a nearest neighbor retrieval mechanism. This model avoids the need for task-specific parameters or finetuning, offering fast and data-efficient adaptation to various scene understanding tasks, including semantic segmentation and depth estimation (at the current stage). The pretraining protocol utilizes a unique combination of attention across and within images, providing robust image representations. Tested on common benchmarks of semantic segmentation and monocular depth estimation, the model's performance approaches the capabilities of specialist models finetuned for each task, suggesting potential for in-context scene understanding.

**Strengths:**

I believe the studied direction, in-context scene understanding, is very important, and this work takes step towards it. The proposed nearest neighbor retrieval framework reminds me of nearest neighbor baselines in few-shot recognition [1,2]. Once the learned representation is powerful enough, a simple NN-baseline can have comparable or even higher performance than SOTA meta-learning algorithm. It is interesting to see such simple framework being re-proposed and shown strong performance in in-context scene understanding.

All the used techniques (cross-attention, patch-level attention, self-contrastive objective) are sound and composed in a reasonable way. Their effectiveness are strongly proven in Table 5. Overall, the paper is easy to follow.





[1] Large-Scale Few-Shot Learning: Knowledge Transfer With Class Hierarchy
[2] SimpleShot: Revisiting Nearest-Neighbor Classification for Few-Shot Learning


**Weaknesses:**

First, I am unsure if Figure 1 is the best design: (1) I cannot know the components used in Hummingbird; (2) does the nearest neighbors (on the top) refers to image-level or patch-level or pixel level? (3) does query image is people-horse the nearest neighbor also contains furniture images?

L 142-144 said: "we train representations to locate the most distinctive part of an image". Does this claim has any supportive evidence?

L 251-254 shows the author adopts a NN search way to let cross-attention to a small set. What's the relationship between the NN search time and memory length.

What's the cost for the pre-training stage? Will this be a huge commitment?

(1) If Hummingbird pre-training with semantic segmentation and fast adopt to monocular depth estimation, how will performance change? (2) If Hummingbird pre-training with monocular depth estimation and fast adopt to semantic segmentation, how will performance change? (3) if Hummingbird pre-training with both semantic segmentation and monocular depth estimation, will this model outperforms than the models pre-trained with one type of data?

**Questions:**

Please address concerns raised above.

**Limitations:**

The author has discussed broader impact and limitations in Section 7 (L 273~285). I personally feel one potential limitation was not discussed: training time and inference time. The training/inference time could be huge if adopting large memory. Also, what's the relationship between training/inference time and memory size?

---

> ### Author Rebuttal · Authors · 2023-08-09
>
> We thank the reviewer for their thoughtful comments.
>
> Weaknesses
>
> * (1) Thank you for the suggestion, we will add a figure showing model components of Hummingbird in the final version. (2) Nearest neighbors are patch-level, we will edit the figure to clarify. (3) The input prompt images are not curated and they contain various kinds of images (including e.g. horses, people, furniture, planes, buildings, etc.) and each patch from the query image attends over patches of all those images to select the nearest neighbors. In the case of the person on a horse query image, its nearest neighbors per the Hummingbird model are the patches from images of people on horses and not the furniture images. In other words, each patch of the query image (e.g. a patch of a horse) will have a similarity score with every patch in the memory bank (which contains many patches, some of horses, some of furniture, some of other things), and the label assigned to a horse patch in the query image will be the weighted average of the patches it is closest to in the memory bank.
>
> * L142-144 This claim follows naturally from the contrastive objective, which learns representations which maximally distinguish an example from others in the batch. When equipped with a spatial attention mechanism, this representation can in particular learn to select the most unique part of the image. This property has been found empirically in [50], which we use as the basis for our spatial attention pooling mechanism. Indeed in [50], the learned attention masks are shown to attend to the main object in an image.
>
> * L251-254 We've attached a graph showing the overhead of the nearest neighbors lookup vs. the length of the memory bank on an A100 GPU in the attached 1-page PDF. For small-to-medium sized memory banks (0 to 1 million keys), the look-up cost is minimal (<= 30ms). When scaling to very large memory banks (10 million+ keys), the scaling tends to be linear (each doubling of the memory bank size resulting in roughly a doubling of the inference time). However, the absolute performance is still decent: with a memory bank size of 10 million, the overhead from NN lookup is only .2 seconds for dense tasks, likely still suitable for most applications. One useful property of the nearest neighbors libraries is that they provide many options to trade-off precision of the NN search with inference time. In this paper we tuned the system for the best performance on the downstream tasks as .2 seconds per image was deemed fast enough for our purposes.
>
> * Cost of pre-training stage: The cost is similar to other contrastive pretraining frameworks. As mentioned in Section 5, contextual pretraining adds a relatively small computational overhead to training the representation (e.g. +22% for $|\mathcal{M}_p| $= 40k relative to standard contrastive pretraining). Total computational requirements for the experiments are reported in section C.2 of the supplementary material.
>
> * We note that the focus of our work was _in-context_ learning of novel scene understanding tasks, which is why we didn’t investigate training on the downstream tasks themselves. All of our models are trained on ImageNet-1k / - 22k and adapted without any change in the parameters to semantic segmentation and depth prediction. Nevertheless this is an interesting question which has partially been addressed in concurrent work (e.g. Visual Token Matching [40]) which showed that such transfer across tasks is indeed possible. Our framework could straightforwardly be adapted to this multitask setting, by framing all downstream tasks as memory retrieval during pretraining rather than during evaluation only.
>
> Limitations
>
> This is an important question, and we will add the following discussion to the appendix. We've attached a graph showing the overhead of the nearest neighbors lookup vs. the length of the memory bank on an A100 GPU in the attached 1-page PDF and we will also add it to the appendix of the final version. The graph shows that the inference time concerns are not as large as one might fear - even with a huge memory bank size of 20 million (far larger than the ideal few-shot/in-context setup), the inference time is less than half a second. For reasonably sized memory banks, even up to 1 million, the inference cost of the NN look-up is nearly free (less than 30 ms per image, making it fast enough to apply to tasks like video processing at 30 frames per second).
>
> The added overhead for contextual pretraining is similarly manageable. Figure 4 shows that the optimal pretraining memory bank size is much smaller for pretraining (O(10^4)) than evaluation (O(10^6)). It is important to note that these two numbers (pretraining memory bank length and inference time memory bank length) have no relationship to one another, meaning we can pretrain with a memory bank length of X and evaluate with a memory bank length of Y. In Section 5, we show that contextual pretraining adds only a relatively small computational overhead to training the representation (e.g. +22% runtime for $|\mathcal{M}_p|$ = 40k relative to standard contrastive pretraining).
>
> It is also important to note that after the one-time cost of pretraining, Hummingbird can be applied to different downstream tasks without any finetuning, which saves on the computation needed to perform inference on a diverse set of tasks.

---

> > ### Comment · Reviewer_KiUe · 2023-08-14
> > **Reviewer response**
> >
> > I thank the rebuttal. I encourage the author to include all the details required at this review into their revision. Also, refer to [50] is not strong enough to support your claims. As you also adopted the contrastive objective, you should can prove via qualitative results if the claim holds.

---

> > > ### Author Response · Authors · 2023-08-15
> > > **Response to Reviewer KiUe**
> > >
> > > Thank you for your response, we will include the text above in the paper revision.
> > >
> > > Following your suggestion, we have visualized the attention masks learned by Hummingbird and observed that they indeed locate the most distinctive part of an image. Since we can't add figures to OpenReview at this stage of the review process, we will include this visualization in the revised version of the paper.
> > >
> > > Please let us know if you have any additional questions or concerns.

---

### Official Review · Reviewer_hsMp · 2023-07-06

**Soundness:** 3 good
**Presentation:** 4 excellent
**Contribution:** 3 good
**Rating:** 8
**Confidence:** 4

**Summary:**

This paper presents a novel in-context learning paradigm based on NN retrieval, which can be adapted for downstream dense vision tasks without requiring fine-tuning or specialized decoders.

**Strengths:**

1. The proposed method addresses a promising regime for reducing task-specific training requirements with prompt images and labels.
2. The framework is conceptually simple yet powerful, providing a concise and reasonable solution.
3. The methodology is clearly presented, and the paper is well-organized and easy to follow.

**Weaknesses:**

1. The current paradigm seems to only be applicable to pixel-wise prediction tasks. As it is unclear how the in-context learning paradigm could be extended to other vision tasks, its application may be limited.
2. The retrieval process is inefficient compared to previous paradigms during inference.

**Questions:**

1. In Tab. 3, why does the result of MoCo-v3 significantly drop for E2E FT?
2. In Sec. 3.2, why contextual pretraining can learn meaningful representations while "enforcing its representation to be expressed as a combination of representations of nearby examples", even without supervision? I don't understand its intuition.

**Limitations:**

The authors have discussed the limitations.

---

> ### Author Rebuttal · Authors · 2023-08-09
>
> We thank the reviewer for their thoughtful comments.
>
> Weaknesses
>
> 1. We focused on the pixel-level dense tasks in this paper due to their relative novelty in the nearest neighbors domain, however, the framework can be used in a straightforward manner for other tasks. For instance, image-level tasks such as classification are very easy to support by pooling the patch-level features for a prompt image and storing this pooled key in the memory bank. The query image can then pool its features and query the memory bank to derive its correct label at the image (rather than pixel/patch) level. For other use cases, nearest neighbor evaluation has historically been used for a wide variety of tasks, such as shape matching [6, 7, 56], scene recognition [62, 71], and image parsing [43, 47]. We discuss some of these examples in the Related Work section of the paper and leave the exploration of these other tasks with our model for future work.
>
> 2. We find that thanks to efficient open-source approximate nearest-neighbor search libraries, we are able to keep the additional inference time necessitated by our approach to a minimum. We've attached a graph showing the overhead of the nearest neighbors lookup vs. the length of the memory bank on an A100 GPU in the attached 1-page PDF and will add this graph to the appendix of the final version. For small-to-medium sized memory banks (0 to 1 million keys), the look-up cost is minimal (<= 30ms), meaning the system is still fast enough to be used for real-time applications, e.g. segmenting videos at 30 frames per second. When scaling to very large memory banks (10 million+ keys), the scaling tends to be linear (each doubling of the memory bank size results in roughly a doubling of the inference time). However, the absolute performance is still reasonable: with a memory bank size of 10 million, the overhead from NN lookup is only 0.2 seconds for dense tasks, likely still suitable for most applications. One useful property of the nearest neighbors libraries is that they provide many options to trade-off precision of the NN search with inference time. In this paper we tuned the system for the best performance on the downstream tasks as .2 seconds per image was deemed fast enough for our purposes.
>
>
> Questions
>
> 1. In this “Fast adaptation” setting, the model is only allowed one pass over the downstream dataset during finetuning, equivalent to 1 epoch. This is different from the typical end-to-end fine-tuning regime in which the model performs multiple passes over the dataset. Note that different models generalize differently to this regime: MAE and MoCo-v3 are slower to adapt, whereas DINO is more robust, despite MAE and MoCo-v3 faring better when given a full compute budget. The fact that different models perform better at slow vs. fast adaptation constitutes an interesting research finding which we believe warrants further investigation.
>
> 2. NN evaluation makes predictions as linear combinations of the labels of nearest neighbor examples. Contextual pretraining aligns the representations with this kind of evaluation by expressing an example’s representation as a linear combination of the representations of its nearest neighbors. By forcing a proportion (determined by the parameter $\lambda$ in L133) of the representation to come from the nearest neighbors before making a prediction, the network learns representations that aren't just good at solving the self-supervised objective (Eq. 4), but good at doing so by attending to their neighbors.

---

> > ### Comment · Reviewer_hsMp · 2023-08-20
> >
> > The response has covered all my concerns and I've raised my rating.

---

### Official Review · Reviewer_32Tr · 2023-07-07

**Soundness:** 3 good
**Presentation:** 4 excellent
**Contribution:** 3 good
**Rating:** 6
**Confidence:** 4

**Summary:**

The paper presents a non-parametric nearest neighbor retrieval method designed to efficiently adapt a pretrained model for dense scene understanding tasks. The fundamental elements include a self-supervised pretraining method, which incorporates attention across images, and a spatial attention-pooling mechanism. The conducted experiments demonstrate improved data efficiency and training speed compared to fully-finetuned methods, while maintaining competitive performance.

**Strengths:**

- The paper shows that non-parametric nearest neighbor retrieval method can be more effective than linear probing in existing self-supervised learning approaches.
- The self-supervised pretraining does not necessitate either fine-tuning or specialized decoders.
- The proposed method significantly outperforms existing self-supervised baselines in downstream tasks and demonstrates improved data efficiency.

**Weaknesses:**

- The paper only consider two downstream tasks (semantic segmentation and depth estimation) in the experiments, despite aiming for general-purpose representations.
- When dealing with a complex dataset like ADE20K, the training approach without any supervision (Hummingbird) exhibits lower performance than some baselines, particularly when the data is scarce (n=158, 316) or when fine-tuning is applied.

**Questions:**

- Why does the proposed contrastive learning objective enable the encoded features to be directly applied to downstream tasks without any parameter adaptation?
This seems to align well with semantic tasks like semantic segmentation, but can you explain its applicability to geometric tasks like depth estimation?

**Limitations:**

The authors have adequately addressed the limitations and broader impact.

---

> ### Author Rebuttal · Authors · 2023-08-09
>
> We thank the reviewer for their thoughtful comments.
>
> Weaknesses
>
> * We demonstrate the generality of Hummingbird representations by evaluating the model on two different tasks (semantic segmentation and depth prediction), of which one is a pixel-level classification task and the other is a regression task. We agree with the reviewer that evaluating on other downstream tasks would be interesting and valuable, and leave this investigation to future work.
>
> *
>   * Low-data regime: In the low-data regime, we compare Hummingbird equipped with the general-purpose nearest neighbor decoder to fully finetuned baselines which can specialize to the task at hand. This puts these baselines at an advantage, particularly for complex datasets such as ADE20K. Despite this disadvantage, Hummingbird is only worse than one baseline (LOCA) in one setting (n=316), while outperforming it in the more challenging n=158 setting, and outperforms it convincingly in all settings on PASCAL (see Table 2).
>
>   * Full fine-tuning: As mentioned in the introduction (L32), the focus of this work is fast and effortless adaptation to downstream tasks (as these are the properties needed for "in-context" scene understanding). In this domain, Hummingbird is indeed superior to prior work. Since fully-finetuned performance wasn’t the focus of the work, we didn't spend as much time tuning the pretraining or fine-tuning hyperparameters here for every variant of Hummingbird, though this would likely help explain/close the gap. We agree that further improving fully-finetuned self-supervised Hummingbird's performance is an interesting avenue for future work as it would allow us to have a single framework be state-of-the-art across many disparate regimes. It is again noteworthy though that the weakest Hummingbird model still outperforms the rest on PASCAL.
>
> Questions
>
> * Contrastive learning in general has indeed been shown to yield semantically relevant features (see e.g. SimCLR [1] and variants thereof), and our use of spatial attention pooling makes these features locally discriminative, preparing them for local semantic tasks such as semantic segmentation. The contextual pretraining of Section 3.2 also helps push the representations to have semantically meaningful nearest neighbors. Moreover, semantically relevant features have been shown to be relevant for geometric tasks such as depth prediction (see e.g. [2]) which has been confirmed by multiple works in SSL (e.g. [3,4]) as well as our empirical results.
>
> [1] T. Chen, S. Kornblith, M. Norouzi, and G. Hinton. A simple framework for contrastive learning of visual representations. In International Conference on Machine Learning, 2020.
>
> [2] ​​Hickson, S., Raveendran, K., Fathi, A., Murphy, K. and Essa, I. Floors are flat: Leveraging semantics for real-time surface normal prediction. In Proceedings of the IEEE/CVF International Conference on Computer Vision, 2019.
>
> [3] C Doersch, A Gupta, AA Efros. Unsupervised Visual Representation Learning by Context Prediction.  Proceedings of the IEEE International Conference on Computer Vision, 2015.
>
> [4] Grill, J. B., Strub, F., Altché, F., Tallec, C., Richemond, P., Buchatskaya, E., ... & Valko, M. Bootstrap your own latent - A new approach to self-supervised learning. Advances in Neural Information Processing Systems, 2020.

---

### Official Review · Reviewer_UEmV · 2023-07-09

**Soundness:** 4 excellent
**Presentation:** 3 good
**Contribution:** 3 good
**Rating:** 8
**Confidence:** 4

**Summary:**

This submission proposed a new representation and pre-training method, named Hummingbird for its fast adaption ability. Hummingbird could perform inference simply by nearest neighbor retrieval from a prompt of annotated features. Unlike the previous pre-training and then fine-tuning/linear probing protocol, the new representation could predict dense outputs by computing the cross attention (dot product similarity with propagation) between the testing query image and the memory bank. The pre-training process could be fully self-supervised or mixed with class label supervision. The new representation of Hummingbird outperforms strong representations like DINO, MAE on nearest neighbor evaluation of semantic segmentation and depth estimation benchmarks by a large margin. Hummingbird also shows great data efficiency in low-data regime.

**Strengths:**

1. I like the idea of nearest neighbor propagation very much and I am very delighted that authors make it works. With the nearest neighbor propagation protocol, a single model could adapt to any vision task without fine-tuning. It's also very interesting that Hummingbird uses patches as the memory bank instead of the whole image, which saves a lot of storage and computation.
2. The fast adaption ability of Hummingbird is very surprisingly good. I like Figure 3 very much, and it shows with just 10s on V100, Hummingbird could achieve very good results already, which is nearly real-time.
3. It's very interesting to see the proposed  self-supervised objective works very well with retrieval-based supervised objective. The self-supervised Hummingbird already achieves amazing results and retrieval-based supervised objective could further boost it.

**Weaknesses:**

1. I am a little bit concerned about the scalability of Hummingbird. I truly appreciate that authors provide ViT-L version of Hummingbird and Hummingbird++. But if we are going to scale Hummingbird to ViT-H level, would ImageNet-22k be sufficient enough? Is there any more gain from scaling up the model? Or if we follow the self-supervised objective only, how much gain could we expect if there is more data?
2. Depth results in "Fast adaptation to downstream tasks" and "Fully finetuned scene understanding" sections are missing. I would also be very interesting to see the adaption ability on tasks other than semantic segmentation.

**Questions:**

1. I found that there is no softmax operation in contextual pretraining Eqn 2 but there is softmax in Eqn 3. Is there any explanation for this design?
2. Would the work be open-sourced? I think this work would be highly impactful for the community and it would be quite beneficial if authors would like to release the implementation and checkpoints.
3. Is Hummingbird ViT-L fully fine-tuning yields higher accuracy than ViT-B?
4. Since Hummingbird is very good at nearest neighbor, it would also be interest to see the linear probing results and comparison.
5. It would be interesting to see whether retrieval-based supervised objective works solely without self-supervised objective.

**Limitations:**

Yes.

---

> ### Author Rebuttal · Authors · 2023-08-09
>
> We thank the reviewer for their thoughtful comments.
>
> Weaknesses
> 1. We agree that further scaling Hummingbird (e.g. to ViT-H) would require scaling the dataset in tandem. For this, we could leverage standard extra-large datasets such as ALIGN, and adapt the supervised retrieval objective to the multimodal contrastive setting. Note that our retrieval-based supervised objective straightforwardly applies here: predictions for the textual modality would be linear combinations of textual embeddings of nearest neighbors, and vice-versa for the image modality.
>
> 2. We agree that this would be an interesting set of experiments to run. Unfortunately, for these two sections we relied on an open-source framework to run the experiments for the baselines for DINO, MAE, MoCo-v3 etc. which had well-tested support for segmentation tasks but not depth related tasks, so collecting this data would require substantial new code beyond our team's current capacity in the rebuttal period.
>
> Questions
>
> 1. If this question refers to why we use mean-pooled hidden representations as opposed to attention-pooled ones as keys of the memory bank: we have experimented with attention pooling, which reduced performance slightly compared to mean pooling. We will add this ablation to the appendix.
>
>     If this question refers to the softmax in cross-attending over the memory bank: the cross-attention operation $\text{CA} (q^i, k_j, v_j)$ (see L132) actually does contain a softmax (please see Eq 1 for the definition of the cross-attention operation) and Eq 2 describes how the keys and values of the memory bank at pretraining time are formed.
>
> 2. We are working on the open source implementation and hope to have it ready by the camera-ready deadline.
>
> 3. Indeed it does. Hummingbird ViT-L full finetuning results weren’t included in Table 4 due to space constraints but we will add the table below in the appendix. We can see that scaling the model size to ViT-L also helps in the finetuning regime, especially when scaling the dataset size accordingly. Note that this scaling study highlights the need for jointly scaling data and model size, in line with the discussion around Weakness #1.
>
> | Model | Backbone | Dataset | PASCAL | ADE20K | Difference ViT-B -> ViT-L
> |:----------------|:-------------:|:-------------:|:-------------:|:-------------:|:-------------:|
> Hummingbird	| ViT-B	 | IN1K 	| 80.0 | 44.9 | |
> Hummingbird	| ViT-L	 | IN1K 	| 82.4 | 47.1 |  +2.4%, +2.2% |
> Hummingbird++ 	| ViT-B	 | IN1K	| 81.2 | 44.9 | |
> Hummingbird++	| ViT-L	 | IN1K 	| 81.8 | 47.3 |  +0.6%, +2.4% |
> Hummingbird 	| ViT-B	 | IN22K 	| 81.6 | 46.9 | |
> Hummingbird 	| ViT-L	 | IN22K 	| 84.1 | 50.8  | +2.5%, +3.9% |
> Hummingbird++ 	| ViT-B	 | IN22K  | 82.1 | 48.2 | |
> Hummingbird++ 	| ViT-L	 | IN22K  | 85.3 | 52.0 | +3.2%, +3.8% |
>
> 4. In Sections 4.2 and 4.3 we compare Hummingbird with nearest neighbor retrieval to Hummingbird with linear probing (i.e. frozen backbone finetuning) and find that retrieval consistently outperforms linear probing. The question then remains how Hummingbird retrieval and Hummingbird linear probing compare to linear probing for other techniques in Table 2 (full nearest neighbor and full fine-tuning comparisons are available in Tables 1 and 4, respectively). To answer this we re-ran the baselines on PASCAL VOC semantic segmentation and found that Hummingbird linear probing far exceeds MAE and MoCo-v3 but slightly lags behind DINO and LOCA in the linear probing regime, though surprisingly, Hummingbird paired with NN retrieval still exceeds all other techniques by a few points, even when they use linear probing rather than NN retrieval.
>
>     This raises an interesting area for future work - more rigorously understanding which properties of the different algorithms lead to good performance given NN decoding vs linear probing vs specialized decoders (e.g. UperNet). Indeed Shi et al. [1] explicitly showed a trade-off between "label efficiency (the ability to learn an accurate classifier on top of the representation with a small amount of labeled data)" and "universality (usefulness across a wide range of downstream tasks)" in the context of contrastive learning with linear probing.
>
> 5. Preliminary experiments ​​with the supervised weight $\alpha$ indicated lower values of $\alpha$ to be necessary for strong downstream task performance (we set $\alpha$ to 0.25 for the supervised ViT-B trained on ImageNet-22k and to 0.05 for all other experiments; see Appendix C.2).
>
>     The ablation below, which is in line with our preliminary findings, shows the results for self-supervised only (Hummingbird), supervised retrieval only, and both objectives (Hummingbird++) for a ViT-B trained on ImageNet-1k and ImageNet-22k. These results highlight the importance of the self-supervised objective in creating general representations that transfer well to downstream tasks. We will include this table in the appendix of the final version.
>
> | Model | Backbone | Dataset | PASCAL | ADE20K | NYUv2 |
> |:----------------|:-------------:|:-------------:|:-------------:|:-------------:|:-------------:|
> Hummingbird (SSL)		| ViT-B	 | IN1K 	| 70.5 | 28.3 | .718 |
> Hummingbird++ (SSL + Sup)	| ViT-B	 | IN1K	| 72.1 | 30.5 | .738 |
> Supervised retrieval only (Sup)	| ViT-B	 | IN1K 	| 56.9 | 22.9 | .787 |
> Hummingbird (SSL)		| ViT-B	 | IN22K 	| 73.5 | 30.7 | .706 |
> Hummingbird++ (SSL + Sup)	| ViT-B	 | IN22K	| 76.2 | 34.1 | .695 |
> Supervised retrieval only (Sup)	| ViT-B	 | IN22K 	| 69.3 | 30.3 | .739 |
>
>
> [1] Z. Shi, J. Chen, K. Li, J. Raghuram, X. Wu, Y. Liang and S. Jha. The Trade-off between Universality and Label Efficiency of Representations from Contrastive Learning. 2023.

---

> > ### Comment · Reviewer_UEmV · 2023-08-20
> > **Response to authors**
> >
> > Thanks for the rebuttal. I like it very much. All my concerns are resolved in the authors' rebuttal. I would keep my score as it is and look forward to your implementation's open source. I think it would inspire the community a lot.

---

### Author Rebuttal · Authors · 2023-08-09

Thank you for your time and effort in reviewing our work.

Please see the attached PDF with a graph showing the overhead of the nearest neighbors lookup vs. the length of the memory bank on an A100 GPU.

---

### Decision · Program_Chairs · 2023-09-21

**Decision:**

Accept (spotlight)

**Comment:**

This paper introduces Hummingbird, a novel representation learning method designed for in-context learning in dense prediction tasks. Hummingbird enhances self-supervised learning by incorporating a spatial attention-pooling mechanism and introducing attention between images, along with an optional retrieval objective during pre-training. During inference, it enables few-shot prediction through retrieval from a limited set of prompted images without any parameter updates. The effectiveness of this method is validated for semantic segmentation and depth estimation.

The paper initially received positive reviews with acceptance recommendations from all reviewers except one. The main concerns pointed out by reviewers related to the scalability of the approach due to retrieval complexity, the applicability of the approach to various scene understanding tasks, and the justifications for pre-training motivation.
The rebuttal effectively addressed these concerns, leading to a consensus to accept the paper after the authors' feedback.
The AC has thoroughly reviewed the submission and the discussions. The AC believes that the method makes a substantial contribution to adapting in-context learning for dense prediction tasks in visual recognition, potentially yielding a significant impact on the community. The proposed retrieval approach for prediction and the adapted pre-training appear to be sound. Consequently, the AC recommends acceptance but strongly advises the authors to include information regarding the scalability and applicability of their approach when preparing the final paper.